# Critical role for the mediodorsal thalamus in permitting rapid reward-guided updating in stochastic reward environments

**Subhojit Chakraborty[1†], Nils Kolling[2], Mark E Walton[2†], Anna S Mitchell[2*†]**

[1]Department of Bioengineering, Imperial College London, London, United Kingdom; [2]Department of Experimental Psychology, Oxford University, Oxford, United Kingdom

**Abstract** Adaptive decision-making uses information gained when exploring alternative options to decide whether to update the current choice strategy. Magnocellular mediodorsal thalamus (MDmc) supports adaptive decision-making, but its causal contribution is not well understood. Monkeys with excitotoxic MDmc damage were tested on probabilistic three-choice decision-making tasks. They could learn and track the changing values in object-reward associations, but they were severely impaired at updating choices after reversals in reward contingencies or when there were multiple options associated with reward. These deficits were not caused by perseveration or insensitivity to negative feedback though. Instead, monkeys with MDmc lesions exhibited an inability to use reward to promote choice repetition after switching to an alternative option due to a diminished influence of recent past choices and the last outcome to guide future behavior. Together, these data suggest MDmc allows for the rapid discovery and persistence with rewarding options, particularly in uncertain or changing environments.

**\*For correspondence:** anna.
mitchell@psy.ox.ac.uk

[†]These authors contributed equally to this work

**Competing interests:** The authors declare that no competing interests exist.

## Introduction

Making adaptive decisions in complex uncertain environments often necessitates sampling the available options to determine their associated values. However, for such exploratory decisions to be of any use for future choice strategies, it is critical that the identity of selected options during 'search' choices are appropriately maintained; without this, the outcomes of such choices can not inform subsequent decisions about whether to continue to sample other alternatives or to terminate the search and instead persist with this chosen option (*Quilodran et al., 2008*). Converging evidence suggests that the integrity of orbital and medial parts of prefrontal cortex supports the ability to use feedback to allow rapid regulation of choice behavior and to shift from search to persist modes of responding (*Hayden et al., 2011*; *Khamassi et al., 2013*; *Morrison et al., 2011*; *Walton et al., 2004*; *2011*). However, it is not yet clear how all the relevant information is efficiently integrated across these cortical networks for this to occur.

One subcortical structure interconnected to these neural networks and therefore in a prime position to help coordinate the rapid integration of choices and outcomes is the mediodorsal thalamus (MD). The MD is heavily interconnected with the prefrontal cortex, and also receives inputs from the amygdala and ventral striatum (*Aggleton and Mishkin, 1984*; *Goldman-Rakic and Porrino, 1985*; *McFarland and Haber, 2002*; *Ray and Price, 1993*; *Russchen et al., 1987*; *Timbie and Barbas, 2015*; *Xiao et al., 2009*). Causal evidence from animal models indicates that MD provides a critical contribution in many reward-guided learning and decision-making tasks, particularly those requiring

**eLife digest** A small structure deep inside the brain, called the mediodorsal thalamus, is a critical part of a brain network that is important for learning new information and making decisions. However, the exact role of this brain area is still not understood, and there is little evidence showing that this area is actually needed to make the best choices.

To explore the role of this area further, Chakraborty et al. trained macaque monkeys to choose between three colorful objects displayed on a touchscreen that was controlled by a computer. Some of their choices resulted in the monkeys getting a tasty food pellet as a reward. However the probability of receiving a reward changed during testing, and in some cases, reversed, meaning that the highest rewarded object was no longer rewarded when chosen and vice versa. While at first the monkeys did not know which choice was the right one, they quickly learned and changed their choices during the test according to which option resulted in them receiving the most reward.

Next, the mediodorsal thalamus in each monkey was damaged and the tests were repeated. Previous research had suggested that such damage might result in animals repeatedly choosing the same option, even though it is clearly the wrong choice. However, Chakraborty et al. showed that it is not as simple as that. Instead monkeys with damage to the mediodorsal thalamus could make different choices but they struggled to use information from their most recent choices to best guide their future behavior. Specifically, the pattern of the monkeys' choices suggests that the mediodorsal thalamus helps to quickly link recent choices that resulted in a reward in order to allow an individual to choose the best option as their next choice.

Further studies are now needed to understand the messages that are relayed between the mediodorsal thalamus and interconnected areas during this rapid linking of recent choices, rewards and upcoming decisions. This will help reveal how these brain areas support normal thought processes and how these processes might be altered in mental health disorders involving learning information and making decisions.

rapid adaptive updating of stimulus values (*Chudasama et al., 2001*; *Corbit et al., 2003*; *Mitchell and Dalrymple-Alford, 2005*; *Mitchell et al., 2007b*; *Mitchell and Gaffan, 2008*; *Ostlund and Balleine, 2008*; *Parnaudeau et al., 2013*; *Wolff et al., 2015*). By contrast, implementation of pre-learned strategies and memory retention remains intact after selective damage to the magnocellular subdivision of MD (MDmc) (*Mitchell et al., 2007a*; *Mitchell and Gaffan, 2008*). Yet the precise role of MDmc in facilitating such rapid learning and adaptive choice behavior remains to be determined.

One potential clue comes from the fact that the functional dissociations occurring after MDmc damage are reminiscent of those observed following lesions to parts of orbitofrontal cortex (OFC) (*Walton et al., 2010*; *Baxter et al., 2007*; *Izquierdo et al., 2004*), to which the MDmc subdivision is reciprocally connected (*Ray and Price, 1993*; *Timbie and Barbas, 2015*). Moreover, intact communication between MDmc and OFC (as well between MDmc and amygdala) is critical for rapid updating of reward-guided choices (*Browning et al., 2015*; *Izquierdo and Murray, 2010*). Lesions to both MD and OFC have been shown to cause deficits on discriminative reversal learning tasks, a finding frequently accompanied by perseveration of choice to the previously rewarded stimulus (*Chudasama et al., 2001*; *Clarke et al., 2008*; *Floresco et al., 1999*; *Hunt and Aggleton, 1998*; *Ouhaz et al., 2015*; *Parnaudeau et al., 2013*; *Chudasama and Robbins, 2003*). This suggests that the main role for MDmc is to promote flexibility by supporting OFC in inhibiting responding to the previously highest value stimulus and/or learning from negative feedback. However, recent functional imaging, electrophysiology and lesion studies have refined theories of OFC function, suggesting it might play an important role in contingent value assignment or in determining the state space to allow such learning to be appropriately credited (*Jocham et al., 2016*; *Walton et al., 2010*; *Takahashi et al., 2011*; *Wilson et al., 2014*). Therefore, a second possibility is that the MD plays a key role in adaptive decision making by facilitating the rapid contingent learning performed by OFC-centered networks.

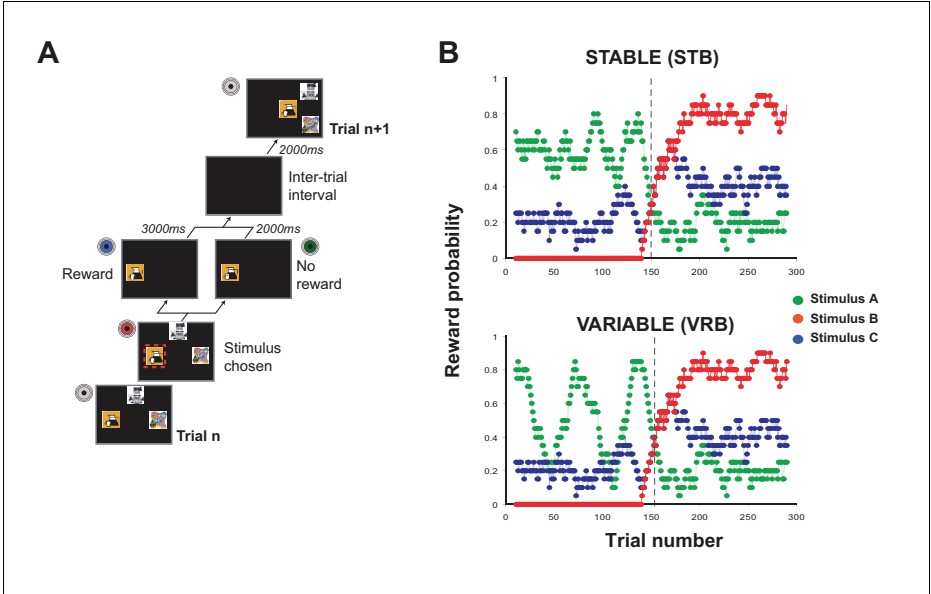

**Figure 1.** Task design. (**A**) Schematic of a single trial. At the start of each trial, 3 stimuli were presented on the screen in one of four spatial configurations. Monkeys chose a stimulus by touching its location on the screen. Once selected, the alternative options disappeared and reward was or was not delivered according to a predetermined schedule (note that the red box is shown for illustration only, but was not presented during testing). Following an intertial interval, the next trial would begin. (**B**) Schematic of two varying schedules, 'Stable' (upper panel) and 'Variable' (lower panel), showing the running average probability (across 20 trials) during a session that selecting that option would result in reward.

However, it is important to keep in mind that causal animal evidence indicates that damage to the MDmc does not simply replicate the deficits observed after selective lesions to interconnected prefrontal regions (*Baxter et al., 2007*; *2008*; *Mitchell et al., 2007a*; *Mitchell and Gaffan, 2008*), suggesting that the functional role of MDmc may be distinct from any individual cortical target. Indeed, in addition to the OFC, the MDmc also has connections to several other parts of rostral, ventral and medial prefrontal cortex (*Goldman-Rakic and Porrino, 1985*; *McFarland and Haber, 2002*; *Ray and Price, 1993*; *Xiao et al., 2009*). These regions are known to be important not just for value learning, but also for aspects of value-guided decision making such as computing the evidence for persisting with a current default option or to switch to an alternative (*Chau et al., 2015*; *Boorman et al., 2013*; *Noonan et al., 2011*; *Kolling et al., 2014*). Based on this evidence, we predicted that the role of MDmc might go beyond simply enabling OFC-dependent contingent learning and might also directly regulate decisions about when to shift from a search strategy (sampling the alternatives to build up a representation of their long-term value) to a persist strategy (repeating a particular stimulus choice).

To determine the precise role of MDmc in facilitating trial-by-trial learning and adaptive decision-making, we tested macaque monkeys before and after bilateral neurotoxic lesions to MDmc, and matched unoperated control monkeys, on a series of probabilistic, multiple option reward-guided learning tasks that are sensitive to OFC damage (*Noonan et al., 2010*; *Walton et al., 2010*). To perform adaptively, the monkeys had to learn about, and track, the values associated with 3 novel stimuli through trial-and-error sampling and use this information to decide whether or not to persist with that option. In some task conditions (referred to as 'Stable' or 'Variable' schedules: see *Figure 1B*), the reward probabilities linked to each stimulus would change dynamically and the identity of the highest value would reverse half way through each session; in others, the probabilistic reward assignments remained fixed throughout the session. If the MDmc is critical for inhibiting responses to a previously rewarded stimulus, then the monkeys with MDmc damage will *only* be impaired post-reversal and will display perseverative patterns of response selection. If, on the other hand, the MDmc supported contingent learning, these lesioned monkeys would show impairments akin to

those observed in monkeys with OFC damage (*Noonan et al., 2010*; *Walton et al., 2010*). That is, MDmc-lesioned monkeys would not only be slower to update their choices post-reversal or, in Fixed situations where they had to integrate across multiple trials to determine which option was best, they would also exhibit aberrant patterns of stimulus choices such that a particular reward would be assigned based on the history of all past choices rather than to its causal antecedent choice. Alternatively and finally, if the MDmc is required to regulate adaptive choice behavior, then the lesioned animals would also have a deficit post-reversal or in any Fixed schedules when multiple options are rewarding, but this would be characterized by an impairment in determining when to shift from search to persist modes of responding.

In fact, while we found that MDmc lesions dramatically influenced the speed and patterns of monkeys' choices, particularly when the identity of the highest rewarded stimulus reversed, there was no evidence either for a failure to inhibit previously rewarded choices or for a misassignment of outcomes based on choice history as had been observed after OFC lesions. Instead, the monkeys with MDmc damage were strikingly deficient at re-selecting a sampled alternative after a search choice that yielded a reward – i.e., they were more likely to select a different stimulus to that chosen on the previous trial. Further analyses suggested this was caused by the MDmc-lesioned monkeys exhibiting less influence of associations based on their most recent stimulus choices coupled with an intact representation of longer term choice trends, which impaired their ability to update their stimulus choices rapidly in situations when they had a varied choice history. Together, these findings support a novel, key contribution of MDmc in regulating adaptive responding.

## Results

A total of ten male rhesus macaque monkeys were trained on a stimulus-guided 3-armed (object) bandit task described in detail in the Procedures and previously (*Noonan et al., 2010*; *Walton et al., 2010*). Briefly, each of the three stimuli was associated with a particular probability of reward according to two predefined schedules (see *Figure 1B*). These particular 'varying' outcome schedules were constructed such that the values of the three stimuli fluctuated continuously throughout the session. However, each schedule was designed to incorporate two properties, namely an initial learning period where one stimulus had an objectively higher value than the other two (here referred to as stimulus 'V1'; note that the only differences between the Stable and Variable schedules was the V1 reward probability in this initial learning period (i.e. during the 1st 150 trials), and a fixed reversal point, after 150 trials, where the identity of V1 changed (*Figure 1B*).

Trial-by-trial reward probabilities were therefore pre-determined and identical for each animal in each comparable session regardless of their choices. The stimuli used in every session were novel, meaning that the monkeys were compelled to learn and track the stimulus values anew through trial-and-error sampling in each test session. The positions of the 3 stimuli on the screen changed on each trial meaning that choices were driven by stimulus identity and not target or spatial location (*Figure 1A*).

Following training and testing on these pre-determined varying outcome schedules, three monkeys then received bilateral neurotoxic (NMDA/ibotenic acid) lesions to MDmc (see Surgery for details) and the other seven remained as unoperated controls. *Figure 2* shows coronal sections of the intended and actual bilateral damage to the MDmc with *Figure 2—figure supplement 1* showing additional coronal sections of the MDmc lesions. All three monkeys (MD1, MD2 and MD3) had bilateral damage to MDmc as intended. Neuronal damage of the MDmc extended throughout the rostral-caudal extent of the nucleus. There was also slight damage to the paraventricular nucleus of the epithalamus positioned directly above the MDmc only in all cases. There was some slight encroachment of the lesion into the parvocellular section of MD3 on the left (see *Figure 2* and *Figure 2—figure supplement 1*). In MD1 and MD2 the parvocellular section remained relatively intact. The more lateral sections of the mediodorsal thalamus remained intact in all three animals. All monkeys also had sagittal section of the splenium of the corpus callosum dorsal to the posterior thalamus. This removal of splenium does not affect performance on other object-reward associative learning tasks (*Parker and Gaffan, 1997*).

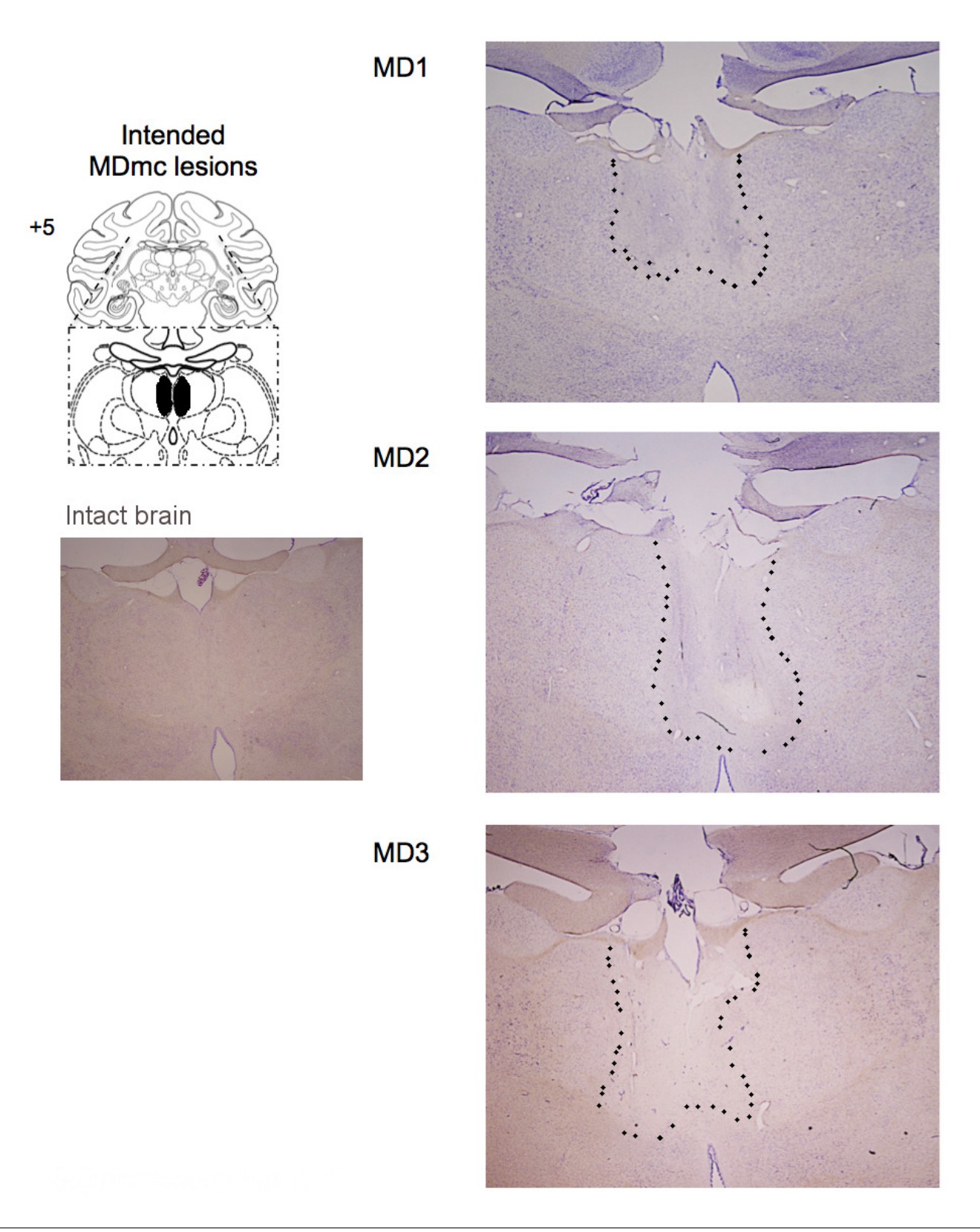

**Figure 2.** Histological reconstruction of the MDmc lesions. Coronal sections (right) corresponding to the schematic diagram (left) with lesion detailed (dotted outline) for the bilateral magnocellular mediodorsal thalamic neurotoxic lesions (MDmc) for the three monkeys, MD1, MD2 and MD3. A corresponding coronal section from an intact monkey has been included for comparison.

The following figure supplement is available for figure 2:

**Figure supplement 1.** Histological documentation of the MDmc lesions.

## Effects of MDmc lesions

Pre-operatively, monkeys in both the control group (n=7) and MDmc group (n=3) were able to rapidly learn to choose the highest value stimulus in either the Stable or Variable schedules (median number of trials to reach criterion of $\geq$65% $V1_{sch}$ choices over 20 trial window: Controls: Stable 25.6 $\pm$ 2.3, Variable 27.7 $\pm$ 6.7; MDmcs: Stable 35.0 $\pm$ 14.0, Variable 21.0 $\pm$ 0.0; all averages are the means across animals $\pm$ S.E.M.) and to update their stimulus choices when the values changed such as after a reversal in the reward contingencies (median number of trials to reach criterion after reversal: Controls: Stable 82.4 $\pm$ 16.0, Variable 79.6 $\pm$ 15.3; MDmcs: Stable 96.3 $\pm$ 28.7, Variable 89.0 $\pm$ 33.9) (*Figure 3*). Comparison of the rates of selection of the best option, either calculated objectively based on the programmed schedules ($V1_{sch}$), or as subjectively defined by the monkeys' experienced reward probabilities based on a Rescorla-Wagner learning algorithm ($V1_{RL}$) (*Figure 3—figure supplement 1*), using a repeated measures ANOVA with lesion group (control or MDmc) as a between-subjects factor and schedule (Stable or Variable) as a within-subjects factor showed no overall difference between the two groups (main effect of group: $F_{1,8}$ < 0.7, p>0.4). The only factor that reached significance was the interaction between group and schedule for the subjectively defined values (Objective values: $F_{1,8}$ = 1.70 p=0.23; Subjective values: $F_{1,8}$ = 5.46, p=0.048). Importantly, post-hoc tests showed that this effect was not driven by a significant difference between the groups on either condition (both p>0.21), but instead by a significant overall difference in performance in the controls between the Stable and Variable that was not present in the MDmc group.

However, after bilateral neurotoxic damage to the MDmc, as shown in *Figure 3*, there was a marked change in choice performance in the MDmc group compared to the control group. A repeated measures ANOVA, with group as a between-subjects factor and both schedule and surgery (pre-MD surgery or post-MD surgery) as within-subjects factors, showed a selective significant interaction of lesion group x surgery for the $V1_{sch}$ ($F_{1,8}$ = 5.537, p=0.046). The interaction of lesion group x surgery for the subjective values ($V1_{RL}$) showed a trend for significance (p=0.054) (*Figure 3—figure supplement 1*). Post-hoc tests indicated that the MDmc group showed a significant decrement in choice performance after surgery, with the lesioned monkeys choosing the highest valued stimulus less frequently than the control monkeys (all p's<0.05). This change was also accompanied by a marked speeding of choice latencies on all trials. An analogous repeated measures ANOVA using the log transformed response latencies showed a significant lesion group x surgery interaction: $F_{1,8}$ = 25.79, p<0.01) (*Figure 3b*).

Previous studies in rodents suggest an important role for MD in reversal learning paradigms (*Block et al., 2007*; *Chudasama et al., 2001*; *Hunt and Aggleton, 1998*; *Ouhaz et al., 2015*; *Parnaudeau et al., 2013*; *2015*), although in these studies the damage sustained in the MD involves all subdivisions of the nucleus. To examine whether our selective MDmc lesion caused a particular problem when needing to switch away from the initial highest valued stimulus ($V1_{sch}$), we separately re-analyzed choice performance during the 1st 150 trials, where the identity of $V1_{sch}$ is fixed, and during the 2nd 150 trials, after the reversal in reward contingencies for $V1_{sch}$ (*Figure 3c*), again including schedule as a within-subjects factor. While the lesion had no consistent effect during the initial learning phase ($V1_{sch}$ 1st half: lesion group x surgery: $F_{1,8}$ = 0.659, p=0.440), there was a significant change in choices post-reversal ($V1_{sch}$ 2nd half: lesion group x surgery interaction: $F_{1,8}$ = 5.990, p=0.040). Post-hoc tests showed that after surgery the MDmc-lesioned monkeys were selectively worse at choosing the $V1_{sch}$ than controls (p=0.031). These data suggest that the monkeys with damage to the MDmc could not flexibly update their choice behavior in a comparable manner to control monkeys following the reversal in identity of the highest value stimulus.

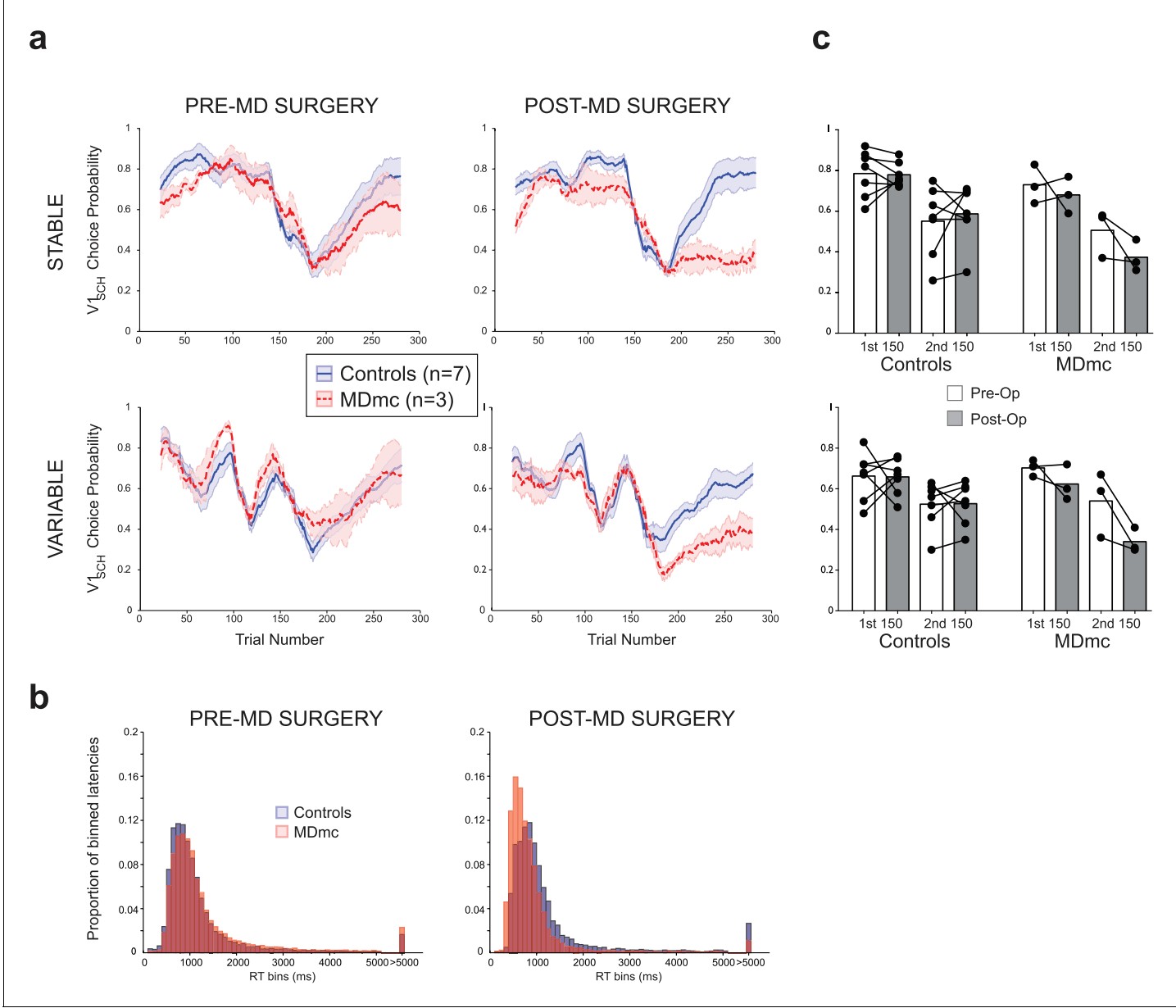

**Figure 3.** Choice performance and latencies on the varying schedules. (a) Mean proportion of choices ( ± S.E.M.) of the $V1_{sch}$ in the control and MDmc groups both pre- and post-operatively. Left and center panel depict group average choices over the whole session (Controls = blue filled line, MDmc = red dashed line); (c) right panel depicts choices divided into the first and last 150 trials (dots = individual monkey's choices). (b) Proportion of trial-by-trial choice response times grouped into 100 ms bins for the controls (blue bars) or MDmc monkeys (red bars).
The following figure supplement is available for figure 3:

**Figure supplement 1.** Choice performance and latencies on the varying schedules for the subjective values.

## MDmc, perseveration and feedback sensitivity

One common explanation for a deficit in behavioral flexibility is that animals with lesions inappropriately continue to choose the previously highest valued stimulus ('perseveration'), potentially because they fail to learn from negative feedback. Such an effect has been observed during reversal learning in rodents following disruption of the MD (*Floresco et al., 1999*; *Hunt and Aggleton, 1998*; *Ouhaz et al., 2015*; *Parnaudeau et al., 2013*). However, despite deficits in reversal learning, our

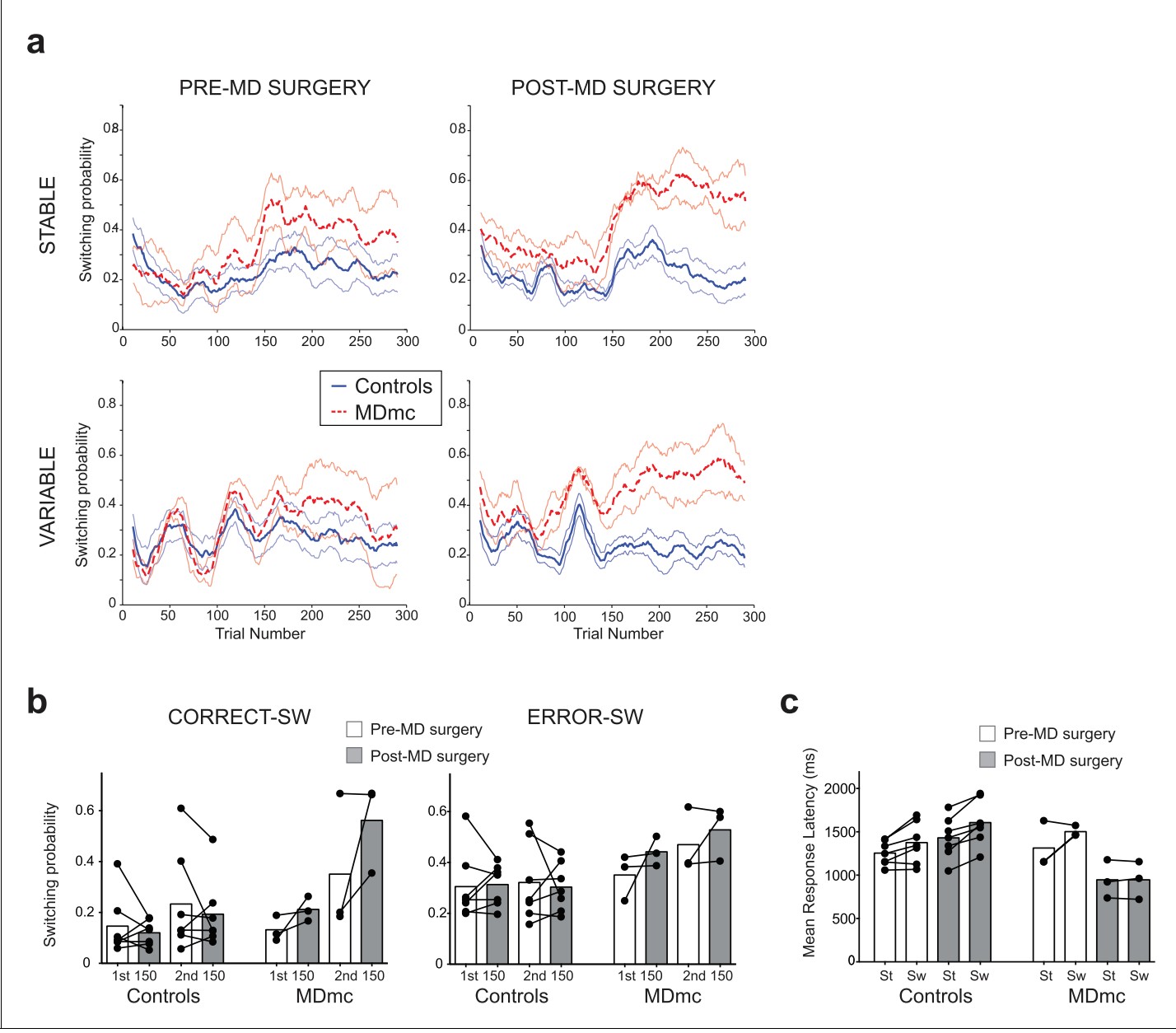

**Figure 4.** Switching behavior in the control and MDmc lesioned monkeys. Mean likelihood of switching to a different stimulus in the two groups both pre- and post-operatively (a) throughout each schedule (mean ± S.E.M.) or (b) divided up into switches (SW) following a choice leading to a reward (CORRECT–SW) or to no reward (ERROR–SW) (dots = individual monkey's switching probabilities). (c) Mean response latency in each animal following a repeated choice of the same stimulus ('St') or a switch to a different stimulus ('Sw') in the two groups (dots = individual monkey's latencies). Note that two MDmc monkeys had very similar latencies pre-operatively and so their data is overlapping.

monkeys with MDmc damage displayed no evidence of either perseveration, or a failure to learn from negative feedback.

First, in the 50 trials after reversal, the MDmc lesion group and the control group had a similar likelihood of choosing what had been the highest valued stimulus pre-reversal (ex-V1$_{sch}$) (proportion of ex-V1$_{sch}$ choices: Controls: Pre-surgery: Stable, 39.9% ± 5.6, Variable, 39.3% ± 3.1, Post-surgery: Stable, 46.7% ± 3.3, Variable, 45.8% ± 6.0; MDmc: Pre-surgery: Stable, 35.3% ± 7.0, Variable, 41.22% ± 2.0, Post-surgery: Stable, 47.1% ± 3.3, Variable, 52.11% ± 1.3; interactions between group x surgery or group x surgery x schedule: both $F_{1,8}$ < 0.6, p>0.45). In fact, as can be observed in

*Figure 4a*, the rate of switching actually *increased* in the MDmc group after surgery. An analysis of the proportion of times the monkeys explored an alternative option after just receiving a reward (positive feedback) or reward being omitted (negative feedback) showed a significant lesion group x surgery x previous outcome interaction ($F_{1,8}$ = 15.01, p=0.005). There was also a strong trend towards a 4-way interaction between lesion group, surgery, previous outcome and pre- or post-reversal ($F_{1,8}$ = 5.12, p=0.054); switch rates selectively increased in the MDmc group after surgery and this was particularly pronounced after a reward (average post-surgery increase in switch probability: after reward 0.15 ± 0.07, post-hoc tests: p=0.051; after no reward: 0.08 ± 0.06, p=0.23).

To explore this increase in switching in more detail, we ran two more repeated measures ANOVAs, one focusing on the pre-reversal period and one on the post-reversal period. While there were no significant interactions with lesion group x surgery in the pre-reversal period (all $F_{1,8}$ < 2.5, p>0.15), there was a significant lesion group x surgery x previous outcome post-reversal ($F_{1,8}$ < 10.9, p=0.011). Further post-hoc tests showed that this effect was mainly driven by a selective increase for the MDmc group to show a tendency to switch to choosing a different stimulus just after having received a reward (increase in switching probability from pre- to post-MD surgery: 0.21 ± 0.07, p=0.056), an effect less evident after no reward (0.06 ± 0.08, p=0.47) or in the control animals after either outcome (reward: –0.04 ± 0.06, p=0.56; no reward 0.02 ± 0.05 p=0.74). This change in switching behavior highlights that, in addition to the absence of evidence for a reduction in sensitivity to negative feedback after an MDmc lesion, there was a change in how positive feedback influenced future choices, particularly in the post-reversal period. As can be seen in *Figure 4b*, this change meant that in the post-reversal phase, monkeys with MDmc damage became no more likely to stay with a current choice after reward delivery than reward omission. In other words, when the identity of the highest value stimulus changed, the MDmc group, postoperatively, was severely impaired at using the receipt of reward as evidence to continue persisting with that particular previously chosen stimulus.

This maladaptive pattern of switching was also reflected in the monkeys' choice latencies. The control monkeys, and MDmc group pre-surgery, all responded slower on trials where they changed their stimulus choice (i.e., choice on current trial 'n' ≠ choice on previous trial 'n-1') compared to trials where they continued to select the same option (choice on trial n = choice on trial n-1) (*Figure 4c*). However, after surgery, the MDmc group failed to exhibit this post-exploration response slowing on exploration trials (group x surgery x switch-stay: $F_{1,8}$ = 5.77, p=0.043) (*Figure 4c*).

## MDmc, contingent learning and the representation of recent choices

MDmc has connections with all parts of the OFC, although they are particularly densest with the lateral OFC, a region that has been implicated in guiding flexible learning and choice behavior (*Murray and Wise, 2010*; *Wallis and Kennerley, 2010*; *Walton et al., 2011*). Therefore, it is possible that impaired contingent value learning, observed after lesions to this region (*Walton et al., 2010*), might also underlie the change in performance observed in the MDmc lesioned group.

One characteristic of the lateral OFC lesion is that, while the monkeys were unable to correctly credit a reward outcome with a particular choice, they still possessed non-contingent learning mechanisms allowing them to approximate value learning based on the weighted history of all recent choices and rewards, irrespective of the precise relationship between these choices and rewards (*Noonan et al., 2010*; *Walton et al., 2010*). This meant that, after a long history of choosing one stimulus (e.g., option A), a new choice (e.g., option B) would be *less likely* to be reselected on the following trial after positive than negative feedback and the previously chosen stimulus A would be *more likely* to be reselected.

To determine whether the MDmc lesioned monkeys also approximated associations based on choice history rather than contingent choice-outcome pairs, we ran a series of analyses to establish the specificity of learning as a function of recent reinforcement and choices. In a first analysis, we looked for the effect described above: whether an outcome – reward or no reward – received for choosing any particular option ('B') might be mis-assigned to proximal choices of another option ('A') as a function of how often 'A' had been chosen in the recent past ('choice history') (note, there were no changes in the 'B' reward likelihood as a function of lesion group, surgery or choice history: all $F$'s < 1.54, p's>0.23). For this analysis, we collapsed across Stable and Variable conditions. As can be observed in *Figure 5a*, pre-operatively both groups were more likely to re-select the previous 'B' option after a reward than no reward across all choice histories. In contrast, post-operatively, the

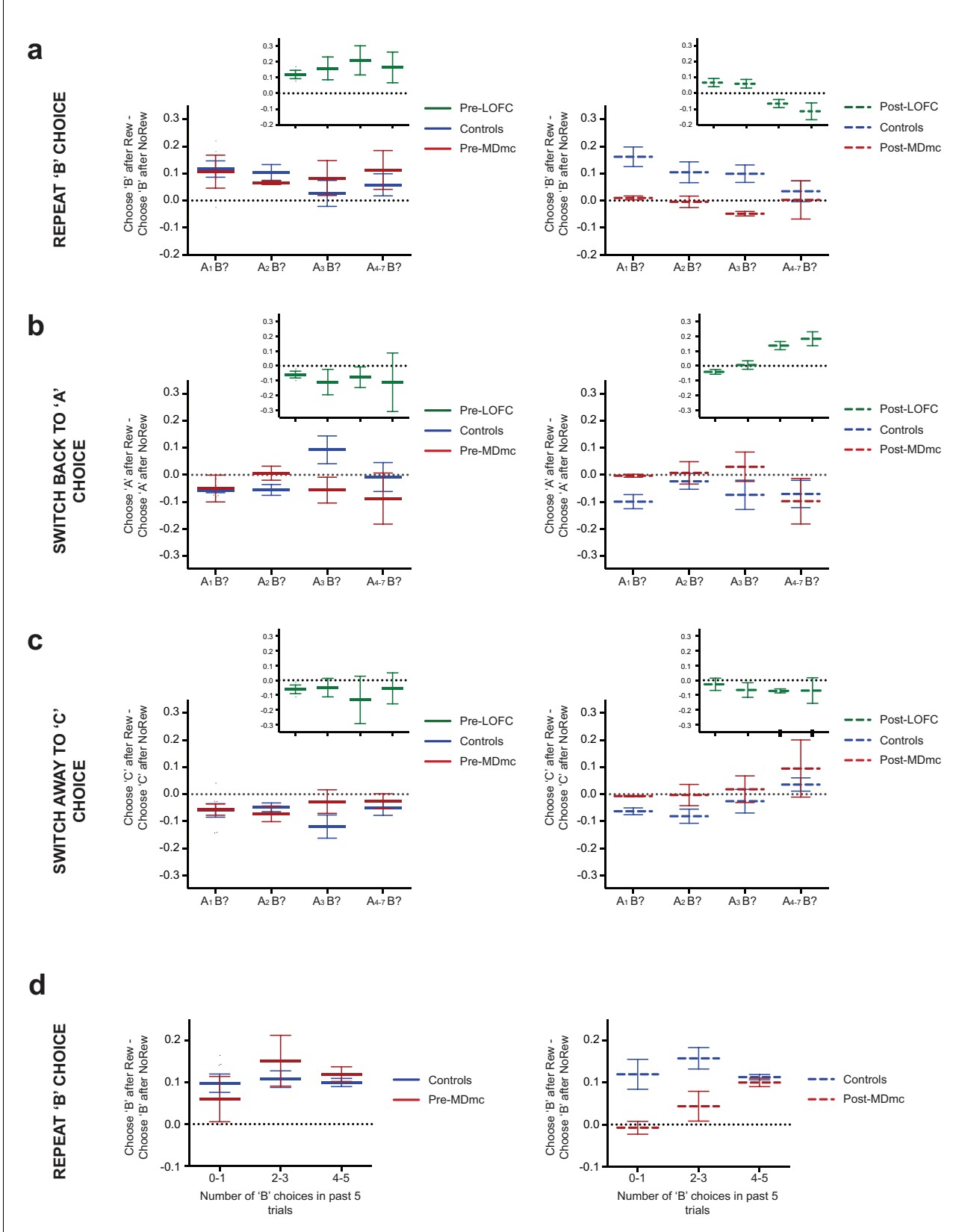

**Figure 5.** Influence of recent choice history over subsequent choices. (**a–c**) Differential likelihood (mean ± S.E.M. across monkeys) of repeating a 'B' choice (**a**), switching back to option 'A' (**b**) or switching away to option 'C' (**c**) after a 'B' choice made on trial n-1 either was rewarded or was not

*Figure 5 continued on next page*

Figure 5 continued

rewarded. Data are plotted in runs following a switch to 'B' as a function of the recent choice history: just one choice of a different 'A' stimulus on trial n–2 ('A$_1$B?'), two choices of 'A' on trials n–2 and n–3 ('A$_2$B?'), three choices of 'A' on trials n–2 to n–4 ('A$_3$B?') or four to seven choices of 'A' on trials n–2 to n–5–8 ('A$_{4-7}$B?'). 'A' and 'B' do not refer to particular stimulus identities but instead to arbitrary choices of one option or another. Main plots show Controls (blue lines) and MDmc (red lines), filled lines = pre-MDmc surgery; dashed lines = post-MDmc surgery. Insets (green lines) depict data from lateral OFC (LOFC) lesioned animals taken from a previous experiment reported in **Walton et al. (2010)**. (d) Differential likelihood (mean ± S.E.M. across monkeys) of repeating a 'B' choice after a reward or no reward plotted as a function of the number of times option 'B' was selected in the 5 previous trials (n–2 to n–6). Controls = blue lines; MDmc = red lines; Pre-MDmc surgery = filled lines; Post-MDmc surgery = dashed lines.

influence of the positive outcome significantly reduced in the MDmc group only (lesion group x surgery interaction: $F_{1,8}$ = 6.32, p=0.036). Further, the number of recent 'A' choices made by the MDmc lesion group did *not* affect the likelihood of re-selecting option 'B'; the MDmc group post-operatively were no less likely to re-select that option after a reward whether they had a short or long choice history on option 'A' (see **Figure 5a**). Moreover, it was not the case that the influence of the recent outcome was being selectively mis-assigned to option 'A' based on recent choices, as was observed in the lateral OFC animals (**Figure 5b**). Instead, the MDmc group showed no overall significant change in their likelihood of reversing back to option 'A' again on the next trial, but rather showed a small increase in the likelihood of switching to the 3$^{rd}$ alternative, 'C' (A choice: group x surgery interaction: $F_{1,8}$ = 2.75, p=0.14; C choice: group x surgery interaction: $F_{1,8}$ = 7.33, p=0.027) (**Figure 5c**). In other words, after a recent switch, the MDmc lesion group was unable to use the past reward as evidence to reselect either their previous choice or even the option chosen most frequently over recent trials.

As can be observed in **Figure 5a–c**, this pattern of choice history appears in marked contrast to monkeys with lateral OFC lesions (**Noonan et al., 2010**; **Walton et al., 2010**). To test this difference formally, we directly compared the groups by re-running the ANOVAs now including the lateral OFC group. The analysis of the likelihood of re-selecting 'B' option again revealed a lesion group x surgery interaction ($F_{2,10}$ = 7.08, p=0.012) but importantly, also now a lesion group x surgery x choice history interaction ($F_{6,30}$ = 2.51, p=0.044). Post-hoc tests showed that, while both the MDmc and lateral OFC groups were on average significantly different to the controls (both p<0.05), the influence of the choice history on the two groups was distinct: only the lateral OFC group post-surgery, but not the MDmc group or controls, exhibited a significant reduction in repetitions of 'B' choices after increasing numbers of previous 'A' choices (p=0.007). Moreover, while the 'B' repetition likelihood was reduced in the MDmc group compared to controls post-surgery, this only occurred when the history of previous 'A' choices was p<0.05 for A$_1$B and A$_3$B, p=0.08 for A$_2$B). In contrast, the lateral OFC group were only different to controls after *3 or more* previous 'A' choices (p<0.05 for A$_3$B and A$_{4-7}$B). Similarly, analysis of the likelihood of returning to option 'A' now revealed a lesion group x surgery interaction, driven by a significant overall increase in 'A' choices in the lateral OFC group that was not present in either the controls or MDmc lesioned animals. Therefore, unlike the lateral OFC lesion group, which displayed less precise and potentially maladaptive *learning* based on associating a past outcome with the history of recent choices, the behavior of the MDmc group was instead characterized by a reduced likelihood of repeating a rewarded choice after just having switched to that option.

Importantly, it was not the case that the MDmc lesioned monkeys were never able to use reward to promote persistence with a chosen option. In a novel companion analysis, we again probed the influence on subsequent behavior of the past choice and outcome, but now investigated how this was influenced by the frequency with which that particular option had been chosen in the previous 5 trials ('choice frequency') (**Figure 5d**). Before surgery, both groups of monkeys were always more likely to persist with an option after being rewarded for that choice than if not rewarded, and this was not significantly influenced by choice frequency. However, post-operatively, although the MDmc lesioned monkeys again were no more likely to re-select the previous choice after reward than after no reward when that option had a low recent choice frequency, this impairment went away in situations when the monkeys had selected that same option on the majority of recent trials. This behavior resulted in a significant surgery x group x choice history interaction ($F_{2,16}$ = 3.80, p=0.045). Therefore, the MDmc lesioned monkeys were just as proficient as controls at weighing the influence of

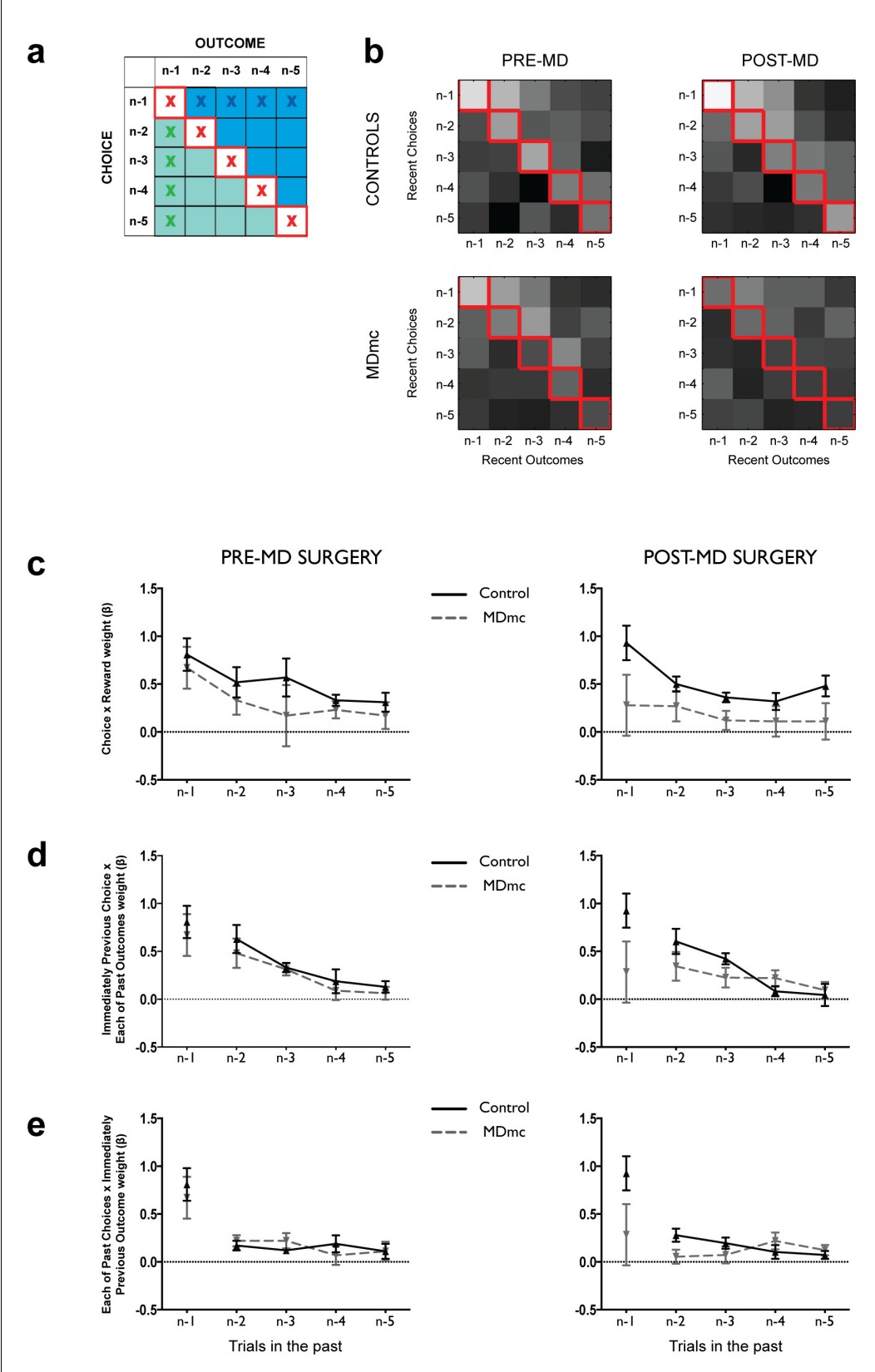

**Figure 6.** Logistic regression on the influence of combinations of recent choices and recent outcomes. (**a**) Representation of the design matrix used in the logistic regression consisting of all combinations of the five previous choices (rows) and five previous outcomes (columns). The white squares on the

*Figure 6 continued on next page*

*Figure 6 continued*

diagonal with red crosses represents the influence of correct contingent learning – choice x outcome combinations; the blue area represents the non-contingent influence of a past outcome spreading forwards to influence more recent choices; the green area represents the non-contingent influence of more recent outcome spreading backwards to associate with an earlier choice. (**b**) Regression weights averaged across the controls and MDmc groups for choices of each of the 3 potential stimuli pre- and post-MD surgery (lighter shades = larger average regression weights; values have been log transformed for ease of visualization). (**c–e**) Regression weights (mean ± S.E.M. across monkeys, arbitrary units) for trials n–1 to n–5 for the contingent choice x outcome pairs (corresponding to the red crosses in a) (**c**), past choice x all previous outcomes (middle panel, blue crosses in panel a) (**d**), and past outcome x all previous choices (lower panel, green crosses in panel a) (**e**).

positive over negative feedback in situations where they had a long choice history on the just chosen option, but not if they had seldom chosen that option in the recent past.

To further investigate the influence of recent choices and outcomes on future behavior, in a third analysis we ran an identical multiple linear regression analysis used previously (*Walton et al., 2010*) focusing on all possible combinations of the past 5 stimulus choices and past 5 outcomes as regressors (all 25 combinations are shown graphically in *Figure 6a*). This analysis allowed us to look not only at how recent specific choice-outcome pairs might guide future behavior (red crosses on *Figure 6a*), characteristic of contingent learning known to depend on the lateral OFC, but also to tease out the influence of false associations between recent choices and unrelated past outcomes (blue area / crosses, *Figure 6a*) or recent outcomes and unrelated past choices (green area / crosses, *Figure 6a*) known not to require an intact lateral OFC. A set of confound regressors from combinations of choices/outcomes 6 trials in the past was also included to capture longer-term choice/reward trends, though not shown in the figures.

Preoperative, both MDmc and control groups exhibited a strong influence of the outcomes they received for the stimuli they had chosen on their future choices, an effect that diminished as the trials became increasingly separated from the current one (*Figure 6b,c*). In other words, they displayed appropriate contingent value learning. Moreover, as had been observed previously, there was also evidence for non-contingent learning mechanisms as demonstrated by a positive influence of the interaction between (i) the most recent choice and unrelated past outcomes (*Figure 6b,d*) and (ii) the most recent outcome and unrelated past choices (*Figure 6b,e*), which also was usually larger for choices / outcomes more proximal to the current trial.

Postoperatively, in the MDmc group, although there was a reduction in the influence of the most recent choice-outcome association, the overall past influence of these specific pairs looking back over a 5 trial history was no different after the lesion to pre-surgery levels (surgery x group interactions: $F$'s < 1.35, p's>0.27) (*Figure 6c*). This lack of effect suggests that stimulus-outcome contingent learning mechanisms do not necessarily depend on the integrity of MDmc. Similarly, there was also no change in the non-contingent association of the previous choice (trial n-1) with unrelated past outcomes (trials n-2 to n-5) (surgery x group interactions: $F$'s < 2.2, p's>0.09) (*Figure 6d*). This result further demonstrates that the MDmc-lesioned animals have intact representations of past outcomes, which can become associated with subsequent choices via 'false' spread-of-effect associations (*Thorndike, 1933*).

By contrast, there was a change in the influence of associations based on interactions between each received outcome and the recent history of choices in the MDmc lesioned monkeys (*Figure 6e*), which resulted in a significant surgery x group x past choice interaction ($F_{4,32}$ = 3.13, p=0.028). To explore this effect further, we re-analyzed the data divided into either more recent choices (stimuli chosen on trial n-1 and n-2) or more distant past trials (choices n-4 and n-5) interacting with the current reward. This analysis revealed a surgery x group x choice recency interaction ($F_{1,8}$ = 9.83, p=0.014). Post-hoc pairwise comparisons demonstrated that the MDmc group, post-surgery, had a significantly diminished influence from associations made between the previous outcome and the most recent contingent and non-contingent choices (p<0.05) although not with more distant non-contingent choices (p>0.2). The interaction remained significant even if the analysis was re-run with recent trials restricted to non-contingent choices on trials n-2 and n-3 to avoid the potential confound that the weight assigned to trial n-1 could result from either correct contingent learning and non-contingent spread-of-effect (surgery x group x choice recency interaction: $F_{1,8}$ = 5.99, p=0.040).

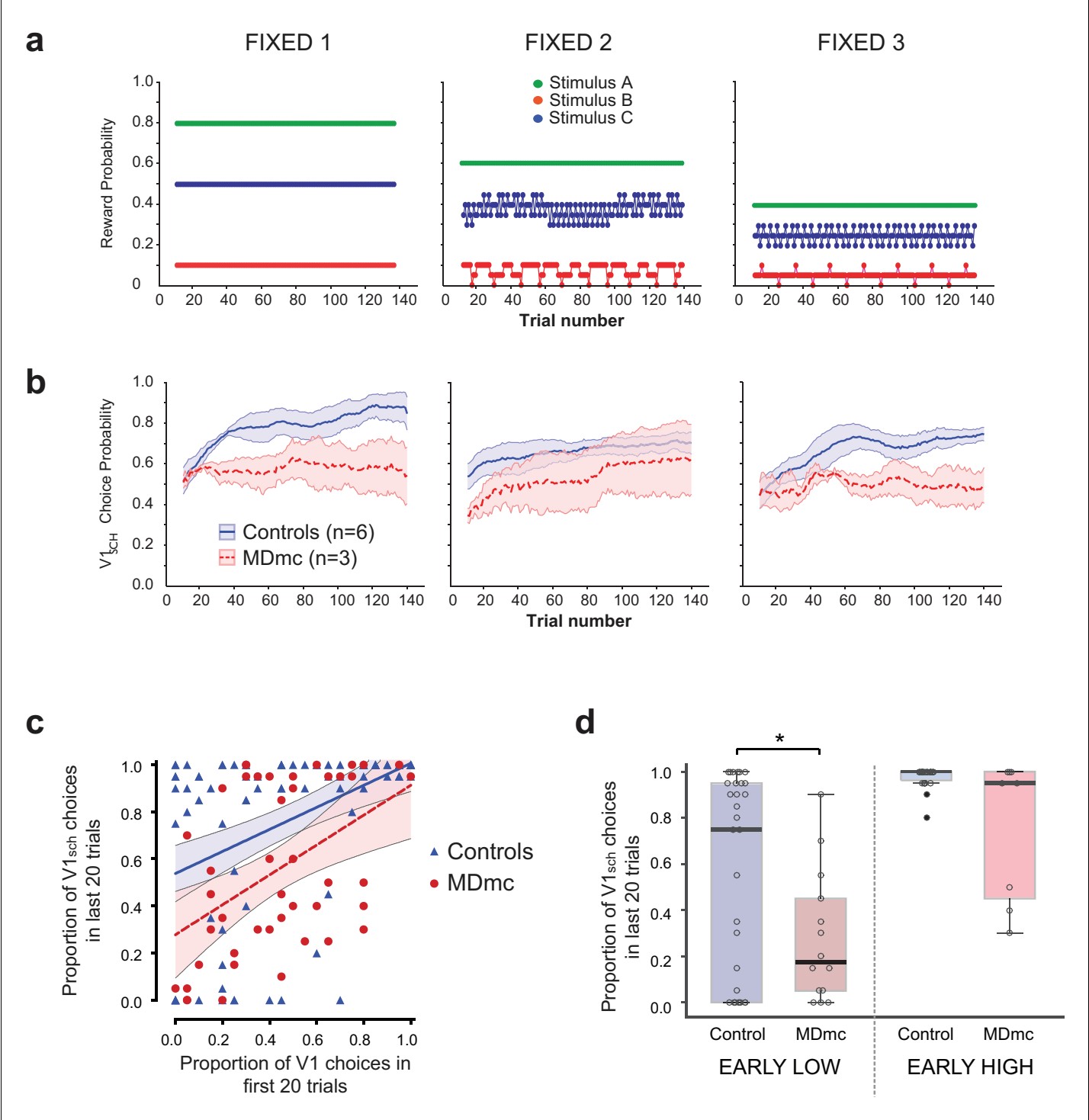

**Figure 7.** Fixed schedule performance. (**a–b**) Schematic of Fixed schedules (upper panels, **a**) and average proportion of choices ( ± S.E.M.) of the V1sch in the control and MDmc groups in each schedule (lower panels, **b**). (**c**) Proportion of V1sch choices in the first and last 20 trials in each session for each animal, plotted along with the best-fit linear regression and 95% confidence limits for each group (Controls or MDmc). (**d**) Box plots showing average proportion V1sch choices in the last 20 trials for sessions in which animals made a low number of V1sch choices in the first 20 trials (≤25% V1sch choices; 'EARLY LOW') or a high number of V1sch choices in the first 20 trials (≥75% V1sch choices; 'EARLY HIGH'). For all box plots, the central mark is the median, the edges of the box are the 25th and 75th percentiles, and the whiskers extend to the most extreme data points not treated as outliers. (*p<0.05, Independent Samples Kolmogorov-Smirnov Test, treating each session as an independent sample.)

Together this significant interaction suggests that the MDmc monkeys do not simply have a primary deficit in contingent value learning. Instead, MDmc monkeys more generally have a degraded representation of their recent – but not more distant – choices, which prevents each outcome from reinforcing the choices made in the past few trials. Without such a mechanism, monkeys will be poor at re-selecting any recently rewarded choices unless they happen to have an extended history of choosing that particular option.

## Performance in fixed reward schedules

Analysis of overall performance on the varying reward schedules (Stable or Variable) highlighted a particular decision making deficit in the MDmc lesioned monkeys that was most prominent when the identity of the best option reversed. However, the problem these monkeys were displaying—a selective failure to persist with a rewarded stimulus choice after having just switched to choosing that stimulus—suggests that the MDmc group may not have a problem with reversals per se but instead in any uncertain situations where they need to use reward to determine which choices should be repeated. This deficit may be particularly pronounced in situations when (a) choice histories are not uniform (e.g., following a reversal or if the value difference between the available options is small) and/or (b) all potential alternatives are associated with some level of reward.

To investigate this idea, we therefore tested the groups post-operatively on additional 3-armed bandit schedules where the reward probability associated with each stimulus was fixed across a session and so there were no reversals of the reward contingencies. In these 'Fixed' schedules (see *Figure 7a*), the reward ratio of the three options remained the same, but the absolute reward yield changed across the schedules (the yield of the second schedule was 0.75 times the first schedule, the third was 0.5 times the first). For all schedules, monkeys have to sample the three stimulus options and use receipt of reward to determine which stimulus to persist with.

The control monkeys rapidly learned to find and persist with the best option in all three schedules, reaching a criterion of choosing $V1_{sch}$ on $\geq 65\%$ of trials on average in $24.4 \pm 5.5$ trials (S.E.M.) across all schedules (note that one control monkey was not run on schedules) (*Figure 7b*). By contrast, even without a reversal, the MDmc lesions affected the rate of learning and likelihood of persisting with $V1_{sch}$, with the MDmc group taking $55.0 \pm 2.2$ trials on average to reach the same criterion. Moreover, as can be observed in *Figure 7b*, the impairment appeared present not just at the start of the session when the animals were initially learning the values, but also persisted throughout the schedule. Therefore, we performed a repeated measures ANOVA comparing performance across Fixed 1–3 ('schedule') on the first and last half of the schedule ('start-end period'). This analysis again revealed a main effect of group ($F_{1,7} = 6.59$, p=0.037), as the MDmc group overall made significantly fewer choices of the best option. Importantly, there was also a significant quadratic interaction between schedule x group x start-end period ($F_{1,7} = 7.01$, p=0.033). Post-hoc tests showed that while the control monkeys made significantly more choices of $V1_{sch}$ in the second half of the period for all three schedules (all $F's_{1,7} > 6.59$, p's<0.05), the MDmc group failed to do this on two out of the three schedules. In other words, across sessions using the three different reward schedules, the MDmc group were consistently impaired at rapidly finding and persisting with the best option in situations when all options had some probability of reward.

Given the results from the varying schedules, we hypothesized that the MDmc lesion should most affect the ability of the monkeys to use reward as evidence to persist with the best option when it had a mixed choice history. Specifically, if the lesioned animals started by mainly sampling the mid and worst options during the initial trials, they should subsequently be less likely to find and persist with the best option; by contrast, if they built up a choice history on the best option in the initial trials, they should then often be just as able as controls to persist with the best option.

To investigate this hypothesis, we examined average proportion of $V1_{sch}$ choices at the start of the session ($1^{st}$ 20 trials) and compared that against performance at the end (last 20 trials) of each of the 5 sessions that the animals completed on the three Fixed schedules. As we had predicted, it was not the case that the MDmc animals never managed to find and persistently select $V1_{sch}$ (defined as choosing $V1_{sch}$ on $\geq 65\%$ of the last 20 trials); this ability occurred on 40% of sessions in the MDmc group (compared to 78% of Fixed sessions in control animals) (*Figure 7c*). Crucially, however, this ability almost never occurred in sessions where they had failed to choose this option on the initial trials (*Figure 7c,d*). To quantify this difference, we contrasted performance at the end of sessions where the animals had either chosen $V1_{sch}$ on $\leq 25\%$ ('EARLY LOW') or $\geq 75\%$ ('EARLY HIGH') of the

first 20 trials. As can be observed in *Figure 7d*, there was a marked difference between the median proportion of $V1_{sch}$ choices at the end of EARLY LOW sessions in the two groups (0.18 $V1_{sch}$ choices for the MDmc group compared to 0.75 for controls; $p<0.05$, Independent Samples Kolmogorov-Smirnov Test, treating each session as an independent sample) but was overlapping in EARLY HIGH sessions (median $V1_{sch}$ choices: 0.95 MDmc group compared to 1.0 for controls; $p>0.05$).

Together, this result demonstrates that the MDmc is not simply required to appropriately update behavior after a reversal, but instead in any situations that require the rapid integration of a reward with a recently sampled alternative to provide evidence for which of several probabilistically rewarded options to persist with.

## Discussion

The current study sought to determine the influence of MDmc when learning and tracking probabilistic reward associations in stochastic reward environments. In the first set of experiments assessing learning and decision-making on the varying reward schedules, we found that the integrity of the MDmc is critical to allow monkeys to update their behavior efficiently following a reversal in the identity of the highest value stimulus. Similar deficits have been previously reported in studies using rats with complete MD lesions (*Block et al., 2007*; *Chudasama et al., 2001*; *Parnaudeau et al., 2013*), an impairment often attributed to failure to prevent perseveration to a previously rewarded option or strategy, though see (*Wolff et al., 2015*). However, such an explanation cannot account for the patterns of choices observed in the current study, as the monkeys with MDmc lesions were no more likely to persevere with the previously highest rewarded option post reversal than controls. In fact, what was most markedly altered in the post-reversal period in the MDmc group was the ability to reselect an option after a rewarded choice of that option. Without this faculty, the lesioned monkeys continued to show maladaptive switching between all three alternatives throughout the post-reversal period and never learned to persist with the new best option (*Figure 4*).

It is not the case, however, that MDmc is simply required whenever there is a need to learn from positive outcomes or to respond on the basis of stimulus identity. The lesioned monkeys were not reliably different from controls in the initial acquisition stage of the varying schedules, despite the use of stochastic reward associations and novel stimuli for each testing session (*Figure 3*). In other studies, similar results of no deficits during acquisition have also been observed. For example, in rodents, complete removal of MD leaves acquisition of serial 2-object visual discrimination learning or 2-choice conditional learning intact (*Chudasama et al., 2001*; *Cross et al., 2012*). Further, in other studies, monkeys with MD or MDmc lesions could acquire concurrent object discriminations when presented across sessions (*Aggleton and Mishkin, 1983*; *Browning et al., 2015*; *Mitchell et al., 2007b*) and could implement a learned decision strategy (*Mitchell et al., 2007a*). Equally, however, it was not that the MDmc is only required to perform appropriately when contingencies reverse. For example, during the Fixed schedules in the current study, the MDmc lesioned monkeys also had a reduced ability to find and persevere with the best option compared to the control monkeys in spite of the fact that the identity of the stimulus associated with the highest reward probability never changed in a session. In the Fixed schedule sessions, the value difference between the options is not substantial and selection of any of the three options could be rewarded, probabilistically.

At first glance, the pattern of results looks very similar to those reported following lesions of the OFC in monkeys (*Walton et al., 2010*), a region heavily interconnected with the MDmc. In that study, the OFC-lesioned monkeys also were initially able to learn and track the value of the best option, but were severely impaired when updating their responses after the identity of the highest value option reversed. OFC-lesioned monkeys also showed deficits on certain fixed schedules. Such a finding might be expected given that the MDmc is the part of MD with major reciprocal connections to the OFC. Indeed, recent behavioral evidence in rodents and monkeys has highlighted that the MD thalamus and cortex work as active partners in cognitive functions (*Browning et al., 2015*; *Cross et al., 2012*; *Parnaudeau et al., 2013*; *2015*).

Nonetheless, our analyses suggest that the two regions play dissociable, though complementary, roles during value-guided learning and adaptive decision-making. The OFC impairment resulted from the loss of an ability to favor associations based on each choice and its contingent outcome, rather than ones based on non-contingent associations between recent history of all choices and all

outcomes. This caused a paradoxical pattern of choice behavior such that the OFC-lesioned monkeys became *more* likely to reselect an option that had been chosen often in the past even if they had just received a reward for selecting an alternative. By contrast, there was no choice history effect in the MDmc lesioned monkeys. In fact, after a recent switch, these monkeys showed no bias towards either the just rewarded option *or* the alternative that had been chosen in the recent past, and instead were more likely to sample the 3$^{rd}$ option on the subsequent trial (*Figure 5a–c*).

This difference in patterns of responding between monkeys with OFC or MDmc damage was also evident in the logistic regression analysis looking at the conjoint influence of the past 5 choices and rewards. Monkeys without an OFC had a selective reduction in the influence of past choice-outcome pairings (*Walton et al., 2010*). In contrast, the MDmc group had a particular loss in the weight assigned to the *most recent past choices* (n-1 to n-3) and the last outcome, but no statistically reliable change across the past trials of precise paired associations between each choice and each outcome. This selective impairment meant that once the monkeys with MDmc damage had an extended choice history on one option (for instance, as occurred on certain sessions at the start of the Fixed schedules: *Figure 7c*), they were just as able as control monkeys to use the outcomes gained from their choices to guide their future behavior. This ability could be seen when examining the monkeys' likelihood of reselecting a stimulus as a function of the number of times that option had been chosen in the past 5 trials (*Figure 5d*). While the controls and MDmc monkeys before surgery exhibited the expected bias to repeat a rewarded choice irrespective of recent history, post surgery the MDmc group only displayed this pattern if they had selected that option on multiple occasions within the recent past. This selective impairment in attributing reward to recent choices, accompanied by the sparing of a faculty to approximate associations based on histories of choices and rewards, is consistent with theories that emphasize the importance of MDmc (and OFC) in goal-directed learning, which requires acquisition of specific future reward predictions of a choice, but not habit learning that relies on longer term trends in choices and outcomes (*Ostlund and Balleine, 2008*; *Bradfield et al., 2013*; *Parnaudeau et al., 2015*).

Taken together, this study implies that a primary function of MDmc is to support the representation of recent stimulus choices to facilitate rapid reward-guided learning and adaptive choice behavior. This function would play a similar role to an eligibility trace in reinforcement learning models, which is essentially a temporary record of recent events used to facilitate learning (*Lee et al., 2012*; *Sutton and Barto, 1998*). Several studies have suggested that MD might be particularly important during rapid task acquisition rather than performance based on previously acquired associations (*Mitchell et al., 2007a*; *2007b*; *Mitchell and Gaffan, 2008*; *Mitchell, 2015*; *Ostlund and Balleine, 2008*; *Ouhaz et al., 2015*). Such a role is also consistent with the electrophysiology finding that some cells in monkey MDmc, as well as in more lateral parvocellular MD, are modulated both when making cue-guided actions and when receiving feedback post-response (*Watanabe and Funahashi, 2004*). The ability to keep track of recent stimulus choices, and their predicted values, is of particular importance when monkeys are sampling alternatives in order to determine the values associated with different objects. At such times, an online representation, or 'hypothesis', of what alternatives *might* be worth sampling would allow rapid updating if their selection leads to a beneficial outcome. Therefore, the MDmc might be described as being critical to facilitate an appropriate balance between exploration and exploitation. However, rather than computing when and what to explore in order to gain valuable new information, functions ascribed to areas such as frontopolar and anterior cingulate cortex and their projecting neuromodulators (*Boorman et al., 2009*; *Donahue et al., 2013*; *Frank et al., 2009*), the role of the MDmc might instead be to help facilitate re-selection and persistence with a beneficial option once it has been found.

In line with this idea, it was notable that one striking effect of the MDmc lesions was that, on top of a general speeding in response latencies, the lesioned monkeys also no longer exhibited a characteristic retardation in latencies on trials where they switch to an alternative compared to when they persisted with the same choice. Taken together with the MDmc lesioned monkeys' increased tendency to sample all three options during exploration, this evidence implies that MDmc is required to exert rapid regulation of stimulus-based choices, particularly when needing to decide when to stop searching and instead persist with a recently sampled optimal option.

# Materials and methods

## Subjects

Subjects were ten rhesus monkeys (*Macaca mulatta*; all males) aged between 4 and 10 years. After preoperative testing, three monkeys received bilateral neurotoxic (NMDA/ibotenic acid) injections under general anesthesia using aseptic neurosurgical conditions (see Surgery details below) to MDmc whereas the rest remained as unoperated controls. Four of these controls were tested alongside the lesioned monkeys. For analysis, the data from these controls were combined with data from three monkeys that were used as unoperated controls in a previously published study using comparable training and identical testing protocols (*Walton et al., 2010*). When the performance of these earlier unoperated controls were compared to the four monkeys tested alongside the lesioned monkeys, they were comparable in performance on all measures, with the exception that the control monkeys from the earlier study selectively made more choices of $V1_{sch}$ in the second half of the Stable schedule (see *Figure 1B*; testing group x condition x session period interaction: $F_{1,7} = 10.14$, p=0.015). Note, however, that the critical statistical tests in this study determine changes *between* the pre- and post-operative testing sessions.

All experimental procedures were performed in compliance with the United Kingdom Animals (Scientific Procedures) Act of 1986. A Home Office (UK) Project License (PPL 30/2678) obtained after review by the University of Oxford Animal Care and Ethical Review committee licensed all procedures. The monkeys were socially housed together in same sex groups of between two and six monkeys. The housing and husbandry were in compliance with the guidelines of the European Directive (2010/63/EU) for the care and use of laboratory animals.

## Apparatus

The computer-controlled test apparatus was identical to that previously described (*Mitchell et al., 2007b*). Briefly, monkeys sat in a transport box fixed to the front of a large touch-sensitive colormonitor that displayed the visual stimuli for all of the experiments. Monkeys reached out through the bars of the transport box to respond on the touchscreen and collect their food reward pellets from a hopper that were automatically dispensed by the computer. Monkeys were monitored remotely via closed circuit cameras and display monitors throughout the testing period.

## Procedures

Prior to the start of the experiments reported here, all monkeys had been trained to use the touchscreens and were experienced at selecting objects on the touchscreen for rewards. On each testing session, monkeys were presented with three novel colorful stimuli, (650 x 650 mm), which they had never previously encountered, assigned to one of the three options (A–C). Stimuli could be presented in one of four spatial configurations (see *Figure 1A*) and each stimulus could occupy any of the three positions specified by the configuration. Configuration and stimulus position was determined randomly on each trial meaning that monkeys were required to use stimulus *identity* rather than action- or spatial-based values to guide their choices. A task programme using Turbo Pascal controlled stimulus presentation, experimental contingencies, and reward delivery.

Reward was delivered stochastically on each option according to predefined schedules. Data are reported from two varying schedules ('Stable' and 'Variable') and three Fixed schedules (*Figures 1b*, *7a*). The monkeys were also tested on several additional varying 3-option schedules, the data from which are not reported here. The likelihood of reward for any option, and for $V1_{sch}$ (the *objectively* highest value stimulus available) and $V1_{RL}$ (the *subjectively* highest value stimulus given the monkeys' choices as derived using a standard Rescola-Wagner learning model with a Boltzmann action selection rule) was calculated using a moving 20 trial window (±10 trials). Whether reward was or was not delivered for selecting one option was entirely independent of the other two alternatives. Available rewards on unchosen alternatives were not held over for subsequent trials. Each animal completed five sessions under each schedule, tested on different days with novel stimuli each time. For the two varying schedules, the sessions were interleaved and data were collected both pre- and postoperatively. For the fixed conditions, the three schedules (*Figure 7a*) were run as consecutive sessions, starting with the five sessions of Fixed 1 (*Figure 7a*, left panel), then five sessions of Fixed 2 (*Figure 7a*, middle panel), and finally five sessions of Fixed 3 (*Figure 7a*, right panel). In all cases for

the reported Fixed schedules, only postoperative data were collected and data acquisition occurred after completion of testing on the varying schedules (note, the animals had performed some other Fixed schedules pre-surgery so had experience of sessions without stimulus reversals). One control monkey was unable to be run on these fixed schedules. The varying schedules comprised of 300 trials per session and the fixed schedules of 150 trials per session.

The data from the varying schedules were analyzed both as a function of $V1_{sch}$ and of $V1_{RL}$. For the latter, a learning rate was fitted individually to each animal's pre-surgery data using standard nonlinear minimization procedures and used for analysis of both pre- *and* post-operative data. Where appropriate, data from all tasks are reported using parametric repeated-measures ANOVA.

The regression analyses were analogous to those described in *Walton et al. (2010)*. In brief, to establish the contribution of choices recently made and rewards recently received on subsequent choices, we performed three separate logistic regression analyses, one for each potential stimulus (A, B, C). For each individual regression, the stimulus in question (e.g., 'A') would take the value of 1 whenever chosen and 0 whenever one of the other two stimuli (e.g., 'B' or 'C') was chosen. We then formed explanatory variables (EVs) based on all possible combinations of recent past choices and recent past rewards (trials n-1, n-2, ..., n-6). Each EV took the value of 1 when, for the particular choice-outcome interaction, the monkey chosen A and was rewarded, –1 when the monkey chose B or C and was rewarded, and the 0 when there was no reward. We then fit a standard logistic regression with these 36 EVs (25 EVs of interest and 11 additional confound regressors describing combinations of choice / outcome n-6). This gave us estimates of $\hat{\beta}_A$ and $\hat{C}_A$.

We then repeated this process for the other two stimuli. This gave us three sets of regression weights, $\hat{\beta}_A, \hat{\beta}_B, \hat{\beta}_C$ and three sets of covariances, $\hat{C}_A, \hat{C}_B, \hat{C}_C$. We proceeded to combine the regression weights into a single weight vector using the variance-weighted mean:

$$\hat{\beta} = \left( \hat{C}_A^{-1} + \hat{C}_B^{-1} + \hat{C}_C^{-1} \right)^{-1} \left( \hat{C}_A^{-1} \hat{\beta}_A + \hat{C}_B^{-1} \hat{\beta}_B + \hat{C}_C^{-1} \hat{\beta}_C \right)$$

## Surgery

Neurosurgical procedures were performed in a dedicated operating theatre under aseptic conditions and aided by an operating microscope. Steroids (methylprednisolone, 20 mg/kg) were given the night before surgery intramuscularly (i.m.), and 4 doses were given 4–6 hr apart (intravenously [i.v.] or i.m.) on the day of surgery to protect against intraoperative edema and postoperative inflammation. Each monkey was sedated on the morning of surgery with both ketamine (10 mg/kg) and xylazine (0.25–0.5 mg/kg, i.m.). Once sedated, the monkey was given atropine (0.05 mg/kg, i.m.) to reduce secretion, antibiotic (amoxicillin, 8.75 mg/kg) as prophylaxis against infection, opioid (buprenorphine 0.01 mg/kg, repeated twice at 4- to 6-hr intervals on the day of surgery, i.v. or i.m.) and nonsteroidal anti-inflammatory (meloxicam, 0.2 mg/kg, i.v.) agents for analgesia, and an H2 receptor antagonist (ranitidine, 1 mg/kg, i.v.) to protect against gastric ulceration as a side effect of the combination of steroid and non-steroidal anti-inflammatory treatment. The head was shaved and an intravenous cannula put in place for intraoperative delivery of fluids (warmed sterile saline drip, 5 ml/h/kg). The monkey was moved into the operating theatre, intubated, placed on sevoflurane anesthesia (1–4%, to effect, in 100% oxygen), and then mechanically ventilated. A hot air blower (Bair Hugger) allowed maintenance of normal body temperature during surgery. Heart rate, oxygen saturation of hemoglobin, mean arterial blood pressure, and tidal CO2, body temperature, and respiration rate were monitored continuously throughout the surgery.

### MDmc lesions

The monkey was placed in a stereotaxic head holder and the head cleaned with alternating antimicrobial scrub and alcohol and draped to allow a midline incision. After opening the skin and underlying galea in layers, a large D-shaped bone flap was created in the cranium over the area of the operation and the dura over the posterior part of the hemisphere was cut and retracted to the midline. Veins draining into the sagittal sinus were cauterized and cut. The hemisphere was retracted with a brain spoon and the splenium of the corpus callosum was cut in the midline with a glass aspirator. The tela choroidea was cauterized at the midline, posterior and dorsal to the thalamus using a metal aspirator that was insulated to the tip. The posterior commissure, the third ventricle posterior to the thalamus and the most posterior 5 mm of the midline thalamus were exposed.

Stereotaxic coordinates were set from the posterior commissure at the midline using the third ventricle as a guide by positioning a stereotaxic manipulator holding a blunt tipped 26-gauge needle of a 10 µl Hamilton syringe above this site. The monkey brain atlas (*Ilinsky and Kultas-Ilinsky, 1987*) was used to calculate the coordinates of the intended lesion site. Neurotoxic bilateral injections to the intended dorsal thalamic nuclei in subjects MD1, MD2 and MD3 were produced by $10 \times 1$ µl injections of a mixture of ibotenic acid (10 mg/ml; Biosearch Technologies, Novato, CA) and NMDA (10 mg/ml; Tocris, Bristol, UK) dissolved in sterile 0.1 mM PBS. This mixture of ibotenic acid and NMDA targets NMDA receptors and metabotropic glutamate receptors has previously produced excellent mediodorsal thalamic lesions in rhesus macaques (*Browning et al., 2015*; *Mitchell et al., 2007a*; *2007b*; *2008*; *Mitchell and Gaffan, 2008*). The needle was positioned for the first set of coordinates: anteroposterior (AP), +5.2 mm anterior to the posterior commissure; mediolateral (ML), ± 1.2 mm lateral to the third ventricle; dorsoventral (DV), −4.0 mm (to compensate for the hole positioned 1 mm above the tip of the needle) ventral to the surface of the thalamus directly above the intended lesion site. Each injection was made slowly over 4 min and the needle was left in place for 4 min before being moved to the next site. The needle was then repositioned for the second set of coordinates: AP, +4.2 mm; ML, ±1.5 mm; DV, −5.0 mm. The third, fourth and fifth sets of coordinates were AP, +4.2 mm, ML, ±1.5 mm, and DV, −3.0 mm; AP, +3.4 mm, ML, ±1.7 mm and DV, −4.0 mm; and AP, +3.4 mm, ML, ±1.7 mm and DV, −3.0 mm, respectively. In each case, the DV coordinate was relative to the surface of the thalamus at the injection site.

When the lesion was complete, the dura was repositioned but not sewn, the bone flap was replaced and held with loose sutures, and the galea and skin were closed with sutures in layers. To reduce cerebral edema, mannitol (20%; a sugar alcohol solution; 1 mg/kg, i.v.) was administered slowly for 30 min while the monkey was still anaesthetized. Then the monkey was removed from the head-holder and anesthesia discontinued. The monkey was extubated when a swallowing reflex was observed, placed in the recovery position in a cage within a quiet, darkened room, and monitored continuously. Normal posture was regained upon waking (waking times varied between 10 and 40 min after the discontinuation of the anesthesia); all monkeys were kept warm with blankets during this time. The morning after surgery, the monkey was moved to a separate cage within their home-room enclosure. Operated monkeys re-joined their socially housed environment as soon as practical after surgery, usually within 3 days of the operation.

After all neurosurgery, each monkey was monitored continuously for at least 48 hr. Postoperative medication continued in consultation with veterinary staff, including steroids (dexamethasone, 1 mg/kg, i.m.) once every 12 hr for four days, then once every 24 hr for three days; analgesia (buprenorphine, 0.01 mg/kg, i.m.) for 48 hr; and antibiotic treatment (amoxicillin, 8.75 mg/kg, oral) for five days. Gastric ulcer protection (omeprazole, 5 mg/kg, oral and antepsin, 500 mg/kg, oral) commenced two days prior to surgery and continued postoperatively for the duration of other prescribed medications, up to 7 days.

## Histology

After completion of all behavioral testing, each monkey was sedated with ketamine (10 mg/kg), deeply anesthetized with intravenous barbiturate and transcardially perfused with 0.9% saline followed by 10% formalin. The brains were extracted and cryoprotected in formalin-sucrose and then sectioned coronally on a freezing microtome at 50 µm thickness. A 1-in-10 series of sections was collected throughout the cerebrum that was expanded to a 1-in-5 series throughout the thalamus. All sections were mounted on gelatin-coated glass microscope slides and stained with cresyl violet.

## Acknowledgements

This work was supported by a Medical Research Council Career Development Fellowship (G0800329) to ASM. MEW was supported by a Wellcome Trust Research Career Development Fellowship (WT090051MA). We wish to thank S Mason for training the monkeys, G Daubney for histology, C Bergmann and Biomedical Services for veterinary and husbandry assistance, MaryAnn Noonan and Tim Behrens for analysis advice, and Matthew Rushworth, Jerome Sallet, Daniel Mitchell and Andrew Bell for helpful discussions about the data.

# Additional information

## Funding

| Funder | Grant reference number | Author |
|---|---|---|
| Medical Research Council | G0800329 | Anna S Mitchell |
| Wellcome Trust | WT090051MA | Mark E Walton |

This work was supported by a Medical Research Council UK Career Development Fellowship (G0800329) to ASM. MEW was supported by a Wellcome Trust Research Career Development Fellowship (WT090051MA).

## Author contributions

SC, Acquisition of data, Drafting or revising the article; NK, Scripted reinforcement learning model, Writing - review and editing, Contributed unpublished essential data or reagents; MEW, Conception and design, Analysis and interpretation of data, Drafting or revising the article; ASM, Conception and design, Acquisition of data, Analysis and interpretation of data, Drafting or revising the article, Contributed unpublished essential data or reagents, performed the neurosurgeries

## Author ORCIDs

Anna S Mitchell, http://orcid.org/0000-0001-8996-1067

## Ethics

Animal experimentation: All experimental procedures were performed in compliance with the United Kingdom Animals (Scientific Procedures) Act of 1986. A Home Office (UK) Project License (PPL 30/2678) obtained after review by the University of Oxford Animal Care and Ethical Review committee licensed all procedures. The monkeys were socially housed together in same sex groups of between two and six monkeys. The housing and husbandry were in compliance with the guidelines of the European Directive (2010/63/EU) for the care and use of laboratory animals. All neurosurgeries were performed under sevoflurane anaesthesia, with appropriate peri-operative medications as advised by our experienced veterinarian, and every effort was made to minimize pain, distress or lasting harm.

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
