## [Decision Letter]

Thank you for submitting your work entitled "Critical Role for the Mediodorsal Thalamus in Strategy Updating during Exploration" for consideration by *eLife*. Your article has been favorably evaluated by David Van Essen (Senior editor) and three reviewers, one of whom is a member of our Board of Reviewing Editors. The reviewers have discussed the reviews with one another and the Reviewing Editor has drafted this decision to help you prepare a revised submission.

This study examined the effects of excitotoxic lesions of the mediodorsal (MD) thalamus on adaptive decision-making in rhesus monkeys performing a battery of probabilistic 3-choice tasks. The experiment follows up on an earlier line of lesion experiments in the OFC that used the same task set. This makes the results of this experiment particularly interesting, because MD provides major thalamic input into OFC, and the behavioral effects of lesion in both areas can be directly compared. In general, lesions of MD and OFC have somewhat similar effects on performance, in both cases degrading the ability to track the best option after a probability reversal. However, the MD-lesioned monkeys appeared to treat recent choice history differently, leading the authors to conclude that the MD group does "not have a primary deficit" in contingency learning but instead has a "degraded representation" of recent choices and recent outcomes.

All three reviewers agreed that the study addresses an interesting and timely topic, was well designed and executed, and produced interesting results. They also all agree on the high value of this kind of combined lesion/behavior study in monkeys to help identify causal contributions of particular brain areas to complex behavior.

However, they also raised a number of serious concerns that will require extensive revisions:

1) The manuscript, as written, does not provide a clear and compelling description of the specific function that MD plays in this kind of decision-making behavior. Among the different descriptions are: i) "facilitate optimal choice stability, particularly in uncertain or changing environments" (stability in what regard – consistent choices? How does this function lead to no effect pre-reversal in the variable and stable conditions?); ii) "allow deliberative control over stimulus-based choices, particularly when in an exploratory mode of responding" (how does more deliberation imply more stability? Shouldn't an effect on deliberation also affect contingency learning?); and iii) "relaying information about current choices to allow frontal-temporal-striatal networks to rapidly integrate and update task relevant strategies on a trial-by-trial basis" (What kind of information? Which "strategies"?). In general, it is not clear how these different claims should be translated into specific predictions about patterns of behavioral deficits that should arise that are specific to the proposed function(s) of MD. The manuscript would be much more compelling if both the specific, hypothesized function of MD, and distinguishable, alternative hypotheses (e.g., behavior becomes more random after an abrupt reversal in reward associations), are described at the beginning, along with their specific predictions. Then the results can be described much more clearly in terms of how well they support or oppose the particular hypotheses.

2) A specific example of this lack of clarity is the description of MD's function in terms of "exploratory" choices. It is not clear how to precisely identify an exploratory choice in the context of the tasks used. It is also not clear why a specific deficit following exploratory choices should lead to the apparently large effects in the fixed condition but none on pre-reversal choices in the stable and variable conditions, since performance in those conditions were described as including "an initial learning period" that presumably includes exploratory choices that, like in the fixed condition, were needed to learn which options were best. After MDmc surgery, were the fixed schedules run before or after the stable/variable schedules? If fixed schedules were run as the first task after surgery, then perhaps the poor performance on fixed relative to first half of stable/variable is due, at least in part, to monkeys learning or relearning to track the best option in conditions with a relatively high yield.

3) The paper's impact depends strongly on the MD lesion results being different from the previously reported OFC lesion results. However, these analyses raise several questions that should be addressed. For example, a key reported difference was the lack of effect of the MD lesion on contingency learning. However, the MD group appeared to have relatively poor contingency learning even before the lesion (Figure 6), suggesting that these animals might not allow such an effect to be identified. Can the authors rule out this possibility? Moreover, the trial-by-trial analyses presented in Figure 5, which are also interpreted as reflecting a "pattern of choice history [that] is in marked contrast to monkeys with lateral OFC lesions," are not the exact same analyses presented in the OFC studies (Noonan et al., 2010, Walton et al., 2010). In addition, for any of these kinds of trial-by-trial choice analyses, were there any differences in the frequency with which the various patterns of choices (e.g., "AB", "AAB", etc., or number of "B" choices in past 5 trials) were rewarded for the different conditions tested (e.g., in control versus lesioned monkeys, who had different patterns of choices and therefore could have, in principle, had "A" and "B" choices for these analyses that were associated with different reward probabilities)? Do any of these choice patterns reflect perseveration with respect to spatial location, not just object identity? In general, it would be useful to directly compare OFC and MD groups on the exact same set of analyses.

4) More task details would be useful to help interpret the behavioral data. For example, in the varying schedules, was the difference between the "stable" and "variable" schedules only in the V1 probability before the reversal point, as seems to be the case in Figure 1? Moreover, when did the reversal point occur (and was it predictable)? What governed the trial-by-trial fluctuations in reward probability in these conditions? Were these fluctuations exactly the same for each experiment? If not, then why did the averaged behavior (e.g., Figure 3, bottom) look so similar to the pattern of reward probability for the schedule shown in Figure 1? If so, then how much did learning across sessions play a role in behavior, both for controls and the lesioned animals? Did the same amount of time elapse between "pre-op" and "post-op" conditions for controls and lesioned animals?

5) The main statistical tests also should be described better and in some cases interpreted more clearly. For example, the figures show data separate by reward schedule (e.g., "stable" versus "variable"), but the text tends to report p values without reference to the specific schedule (e.g., line 144-45: "post-operatively as shown in Figure 3, there was a marked change in choice performance in the MDmc group[…]“). Were data simply combined across schedules? Was schedule type a factor in the ANOVAs? In general, the authors should more clearly explain what they mean by their terms for the factors in their ANOVA. For example, what is the difference between 'surgery' and 'condition'? Moreover, in the Results section it states that "This highlights that, while there was no evidence of a reduction in sensitivity to negative feedback after an MDmc lesion, there was a specific change in how positive feedback influenced future choices in the post-reversal period." However, the statistical test presented just prior to this statement showed that the interaction term including pre- and post-reversal was not significant.

Finally, the regression described in Figure 6 should be described better. What exactly are the "averages" shown in the grids in panel a? across monkeys? Does each square correspond to a single coefficient (if so, how?) or several (this is my guess – one for each pairing of three choices and two outcomes for the given n, correct?)?

---

## [Author Response]

1) The manuscript, as written, does not provide a clear and compelling description of the specific function that MD plays in this kind of decision-making behavior. Among the different descriptions are: i) "facilitate optimal choice stability, particularly in uncertain or changing environments" (stability in what regard – consistent choices? How does this function lead to no effect pre-reversal in the variable and stable conditions?); ii) "allow deliberative control over stimulus-based choices, particularly when in an exploratory mode of responding" (how does more deliberation imply more stability? Shouldn't an effect on deliberation also affect contingency learning?); and iii) "relaying information about current choices to allow frontal-temporal-striatal networks to rapidly integrate and update task relevant strategies on a trial-by-trial basis" (What kind of information? Which "strategies"?). In general, it is not clear how these different claims should be translated into specific predictions about patterns of behavioral deficits that should arise that are specific to the proposed function(s) of MD. The manuscript would be much more compelling if both the specific, hypothesized function of MD, and distinguishable, alternative hypotheses (e.g., behavior becomes more random after an abrupt reversal in reward associations), are described at the beginning, along with their specific predictions. Then the results can be described much more clearly in terms of how well they support or oppose the particular hypotheses.

Thank you for these thoughtful comments. We understand that we were not sufficiently clear in the original manuscript in (a) describing our hypotheses, (b) whether these hypotheses were or were not confirmed by the data, and (c) our functional interpretation of the overall pattern of deficits following the MDmc damage. We have substantially revised the entire manuscript to address these issues. Specifically:

In the Introduction, we have set out 3 potential functions of MDmc during adaptive learning and decision making, namely, (i) updating choices after a reversal through inhibition of responses to a previously rewarded stimulus (and/or through learning from negative outcomes; (ii) enabling OFC-dependent contingent learning mechanisms; (iii) facilitating adaptive shifts from a “search” strategy (i.e., sampling the available options to build up a representation of their long-term value) to a “persist” strategy (repeating a particular stimulus choice).

We next set out what patterns of deficits we might expect to see after the MDmc lesion according to each theory:

“If the MDmc is critical for inhibiting responses to a previously rewarded stimulus, then the monkeys with MDmc damage will *only* be impaired post-reversal and will display perseverative patterns of response selection. […] Alternatively, and finally, if the MDmc is required to regulate adaptive choice behavior, then the lesioned animals would also have a deficit post-reversal or in any Fixed schedules when multiple options are rewarding, but this would be characterized by an impairment in determining when to shift from search to persist modes of responding.”

We have gone through the manuscript to ensure the terminology we use to describe the patterns of impairment are consistent throughout. For instance, we have cut reference to “optimal choice stability” and “deliberative control”. Instead, we focus on (a) the impairment in repeating a rewarded choice following a recent switch in stimulus choices (what we term moving from “search” to a “persist” strategy) and (b) the specific reduction in influence of associations made between the previous outcome and stimuli chosen on the most recent trials (n-1 – n-3) but not more distant choices (trial n-4 – n-5), which reflects a degraded representation of the most recent stimulus choices.

Together, this implies a critical role for MDmc in using reward in uncertain, multi-option environments to bias choices towards choice repetition and away from constant sampling of the different alternatives when searching for the best option available.

2) A specific example of this lack of clarity is the description of MD's function in terms of "exploratory" choices. It is not clear how to precisely identify an exploratory choice in the context of the tasks used. It is also not clear why a specific deficit following exploratory choices should lead to the apparently large effects in the fixed condition but none on pre-reversal choices in the stable and variable conditions, since performance in those conditions were described as including "an initial learning period" that presumably includes exploratory choices that, like in the fixed condition, were needed to learn which options were best. After MDmc surgery, were the fixed schedules run before or after the stable/variable schedules? If fixed schedules were run as the first task after surgery, then perhaps the poor performance on fixed relative to first half of stable/variable is due, at least in part, to monkeys learning or relearning to track the best option in conditions with a relatively high yield.

We apologise for the lack of clarity with our terminology. We had originally chosen the term “exploratory” – now changed to “search” or simply “switch” in the revised manuscript – as the deficit in the MDmc-lesioned animals is specifically characterised by a reduced ability to persist with a rewarded choice on trial N+1 if a new stimulus choice had been made on trial *N* (Figure 4, Figure 5). In other words, following a switch choice, the MDmc-lesioned animals were impaired at using reward as evidence to repeat that choice. However, the lesion spared several important abilities. First, there was no change post-operatively in the likelihood of switching after an unrewarded choice. Second, the lesioned animals were able to persist with a rewarded stimulus choice so long as they had an extended recent history of choosing that option (see Figure 5).

The reviewer therefore raises the key question about what it is that defines a task situation where MDmc lesioned animals were deficient. We can rule out that the difference in between the Fixed schedules and initial learning parts of Stable / Variable resulted from any relearning of a strategy as the Fixed schedules were run after the Stable/Variable schedules. We have stated this in the revised manuscript.

Instead, we believe the pattern of deficits depends strongly on two interrelated factors: (a) the reward probabilities of the available options and (b) the animals’ particular history of choices. Given the lesioned animals’ problem in re-selecting a stimulus after a reward when that stimulus has just been switched to, MDmc impairments should be most prominent in conditions that promote exploration – i.e., following a reversal or where the stimulus values are close together and/or low (the latter situations characterise the Fixed schedules). However, in conditions where only one option is frequently rewarded, the best option will likely be chosen more frequently and therefore MDmc group performance should be relatively unaffected. This is the case in the initial period of the Stable and Variable schedules.

To demonstrate this effect, we re-analysed performance on the Fixed schedules, examining performance at the end of each individual session as a function of best option choices at the start of the session (defined as the proportion of best option choices in the 1^st^ 20 trials). Even though the MDmc group were impaired *on average* at finding and persisting with the best option, we hypothesised that performance in each session would depend strongly on performance in the initial trials. Specifically, if the lesioned animals mainly sampled the mid and worst options during the initial trials, they should subsequently be less likely to find and persist with the best option; by contrast, if they built up a choice history on the best option in the initial trials, they would then perform similarly as controls at persisting with the best option.

As can be seen in Figure 7 in the revised manuscript, this is exactly what we found. The MDmc animals did manage to find and persistently select the best option (defined as choosing V1_sch_ on ≥65% of the last 20 trials) in 40% of all Fixed sessions (compared to 78% of Fixed sessions in control animals). However, this almost never occurred in sessions where they had failed to choose this option on the initial trials (panel A). This is even clearer when examining performance at the end of sessions divided up into those where the animals chose the best option on either ≤25% (“EARLY LOW”) or ≥75% (“EARLY HIGH”) of the first 20 trials (panel B). As can be observed, there was a marked difference between the median proportion of V1_sch_ choices at the end of EARLY LOW sessions in the two groups (0.18 V1_sch_ choices for the MDmc group compared to 0.75 for controls; *p* < 0.05) but was overlapping in EARLY HIGH sessions (median V1_sch_ choices: 0.95 MDmc group compared to 1.0 for controls; *p* > 0.05).

In fact, we did run one Fixed condition, not included in the original manuscript, that had a similar spread of reward probabilities as the initial learning period of Stable (reward probabilities for V1-V3: 0.6: 0.2: 0 for Fixed v 0.61: 0.21: 0 for Stable; see Figure 8 panel A) As can be seen in Figure 8, post-operative Fixed choice performance in the MDmc group align with performance in the 1^st^ half of the Stable condition and overlap with the control group. We chose not to include these data as the performance of one MDmc animal was very different to its performance in all previous sessions associated with the Fixed schedules and also to the other two lesioned animals, which obscures the main finding. However, if the reviewer thinks this would be helpful to have included, we are happy to do so.

Author response image 1.Comparison between choice performance during initial learning period of Stable with an equivalent “Fixed” condition.Reward probabilities for choosing each option during the 1^st^ 150 trials of Stable (**A**) and an equivalent Fixed schedule (**B**) and individual animals’ average proportion of V1_sch_ choices in these schedules (Stable, C; Fixed, D).**DOI:**
http://dx.doi.org/10.7554/eLife.13588.012

We hope that the inclusion of some of the above analyses, as well as the extensive revisions to the Introduction and Results have now clarified these issues.

3) The paper's impact depends strongly on the MD lesion results being different from the previously reported OFC lesion results. However, these analyses raise several questions that should be addressed.

Thank you for highlighting this point. We absolutely agree that the comparison with the previous OFC data is critical here. We will deal with each point raised in turn:

For example, a key reported difference was the lack of effect of the MD lesion on contingency learning. However, the MD group appeared to have relatively poor contingency learning even before the lesion (Figure 6), suggesting that these animals might not allow such an effect to be identified. Can the authors rule out this possibility?

We are confident that this is not an issue. As can be seen in Figure 9, which depicts the contingent learning regression weights for each animal before the lesion, the apparent reduction in the influence of contingent pairings that the reviewer noticed in the MD group pre-surgery was driven by 1 of the 3 animals; the other two animals assigned to the MDmc group exhibited an influence of past choice x outcome pairs equivalent to most other animals in the task. Moreover, *all* monkeys during pre-surgery testing showed a significant positive influence of recent choice x reward pairings that was greatest on trial n-1.

Author response image 2.Influence of contingent recent choice – outcome pairs on the current choice.**DOI:**
http://dx.doi.org/10.7554/eLife.13588.013

Out of the group of 10 monkeys trained on the task, two of them consistently performed slightly worse pre-operatively than the group average (though still above our behavioural criteria for inclusion). To ensure these individuals did not bias the results in either direction, one of these was assigned to the control group and the other to get an MD lesion.

Moreover, the trial-by-trial analyses presented in Figure 5, which are also interpreted as reflecting a "pattern of choice history [that] is in marked contrast to monkeys with lateral OFC lesions," are not the exact same analyses presented in the OFC studies (Noonan et al., 2010, Walton et al., 2010).

While the reviewer is factually correct that the analyses are not the exact same ones as presented in the main figures of the two mentioned papers, it is important to appreciate they are essentially analogous: they examine how the immediate past reinforcement influences the next choice as a function of the recent pattern of choices. The main difference is that instead of presenting the likelihood of switching back to an ‘A’ choice after a ‘B’ choice, we instead presented the likelihood of persisting with that ‘B’ choice on the next trial. Given that we had already observed a tendency to fail to persist following positive feedback, we felt that this was potentially the most informative analysis. These analyses are also slightly different in that data from additional testing schedules were included in the Walton/Noonan 2010 papers.

Nonetheless, we understand the reviewer’s fundamental point and in the revised manuscript have:

A) Presented complementary figures depicting the likelihood of (i) switching back to ‘A’ or (ii) switching to novel option ‘C’ depending on recent choice history and reinforcement on the previous trial (Figure 5).

B) Included the equivalent data from the “lateral OFC” lesion group from Walton et al. 2010 as insets in Figure 5.

C) Performed novel analyses including the lateral OFC group in the ANOVA to determine the separate influence of the MDmc and lateral OFC lesion on how the past reward influences subsequent stimulus choices as a function of the recent choice history.

As now can be directly observed, while the lateral OFC lesioned animals’ choices were strongly influenced by the recent choice history, causing them to be more likely to switch *away* from ‘B’ and back to ‘A’ after a reward on option ‘B’ if they had chosen option A on many trials in the recent past, there was no such influence on the MDmc group’s choices. Instead, these animals exhibited no consistent bias towards repeating a rewarded choice and were also just as likely to switch to option ‘C’ as to return to option ‘A’. In other words, while the lateral OFC lesion group were displaying less precise and potentially maladaptive *learning*, the MDmc lesion group were impaired at exploiting a rewarded choice after just having switched to that option.

In addition, for any of these kinds of trial-by-trial choice analyses, were there any differences in the frequency with which the various patterns of choices (e.g., "AB", "AAB", etc., or number of "B" choices in past 5 trials) were rewarded for the different conditions tested (e.g., in control versus lesioned monkeys, who had different patterns of choices and therefore could have, in principle, had "A" and "B" choices for these analyses that were associated with different reward probabilities)?

No, the likelihood of receiving a reward for a particular ‘B’ choice was unaffected by the recent choice history or by the lesion group (see Figure 10). An analysis of these data comparing reward likelihood after each sequence in the two groups pre- and post-MD surgery found no differences on any measure (all F < 1.54, p > 0.23). We have described this in the revised manuscript:

“(note, there were no changes in the ‘B’ reward likelihood as a function of lesion group, surgery or choice history: all *F*’s < 1.54, *p*’s > 0.23)”

Author response image 3.Probability of reward on the ‘B?’ trial as a function of recent reward history.There were no differences between the groups (all F < 1.54, p > 0.23).**DOI:**
http://dx.doi.org/10.7554/eLife.13588.014

4) More task details would be useful to help interpret the behavioral data. For example, in the varying schedules, was the difference between the "stable" and "variable" schedules only in the V1 probability before the reversal point, as seems to be the case in Figure 1? Moreover, when did the reversal point occur (and was it predictable)? What governed the trial-by-trial fluctuations in reward probability in these conditions? Were these fluctuations exactly the same for each experiment? If not, then why did the averaged behavior (e.g., Figure 3, bottom) look so similar to the pattern of reward probability for the schedule shown in Figure 1? If so, then how much did learning across sessions play a role in behavior, both for controls and the lesioned animals? Did the same amount of time elapse between "pre-op" and "post-op" conditions for controls and lesioned animals?

In the revised manuscript, we have included more details about the task to clarify exactly how the reward schedules were implemented. In order of above:

Yes, the only difference between Stable and Variable pre-reversal was the reward probability of V1.

The reversal point always occurred in these conditions in the same fixed place, irrespective of performance. Note, however, that both pre- and post-surgery, prior to being tested on Stable / Variable, the animals were tested using three separate 3-armed bandit schedules where reversals happened at different points in the session. They had also performed some Fixed schedules pre-surgery (i.e., schedules without a reversal). Therefore, while the animals may have had an expectation of some change in reward probabilities, it is very unlikely that they would have built up a prior expectation of when this would occur.

Trial-by-trial reward schedules were predetermined and fixed across sessions. The schematic of the reward probabilities is based on a 20-trial running average of the reward rate for each option.

Sessions with the Stable and Variable schedules were interleaved over 10 testing sessions. As mentioned above, pre-operative Stable and Variable sessions occurred *after* the animals had experienced 15 sessions of testing on three separate 3-armed bandit schedules, each with distinct reward schedules. All animals only moved onto pre-operative testing following extensive training on (a) simpler probabilistic reversal schedules and (b) having achieved a behavioural criterion on a different 3-armed bandit schedule. Therefore, as far as we could ascertain, there were no consistent changes in performance across sessions in either the controls or lesioned animals (i.e., a significant main effect or interaction with testing session in the analyses driven by a progressive change across sessions).

5) The main statistical tests also should be described better and in some cases interpreted more clearly.

We apologise for this lack of clarity about our analyses. We have revised the manuscript thoroughly to ensure that all tests are described in detail and terminology is standardised. Again, we will deal with each point raised in turn.

For example, the figures show data separate by reward schedule (e.g., "stable" versus "variable"), but the text tends to report p values without reference to the specific schedule (e.g., line 144-45: "post-operatively as shown in Figure 3, there was a marked change in choice performance in the MDmc group[…]“). Were data simply combined across schedules? Was schedule type a factor in the ANOVAs? In general, the authors should more clearly explain what they mean by their terms for the factors in their ANOVA. For example, what is the difference between 'surgery' and 'condition'?

In the revised manuscript, we have endeavoured to detail precisely the factors in each ANOVA. For example:

“Comparison of the rates of selection of the best option, either calculated objectively based on the programmed schedules (V1_sch_), or as subjectively defined by the monkeys’ experienced reward probabilities based on a Rescorla-Wagner learning algorithm (V1_RL_), using a repeated measures ANOVA with lesion group (control or MDmc) as a between-subjects factor and schedule (Stable or Variable) as a within-subjects factor showed no overall difference between the two groups (main effect of group: *F*_1,8_ < 0.7, *p* > 0.4).”

“A repeated measures ANOVA, with group as a between-subjects factor and both schedule and surgery (pre-MD surgery or post-MD surgery) as within-subjects factors, showed a selective significant interaction of lesion group x surgery for the V1_sch_ (*F*_1,8_ = 5.537, *p* = 0.046).”

“Schedule” was a factor in all of the ANOVAs, except for the more fine-grained ‘choice history’ and regression analyses where we pooled across schedules to increase power. In fact, part of the reviewer’s confusion stemmed from the fact that we accidentally sometimes referred to Schedule as “Condition” in the presentation of the ANOVAs in the original submission; “Condition has been changed to “Schedule” throughout in the revised manuscript. In practice, we found virtually no meaningful interactions between the testing schedule in the varying conditions (i.e., Stable v. Variable) and our effects of interest (i.e., lesion group and surgery); any that did reach significance are now explicitly stated in the text.

Moreover, in the Results section it states that "This highlights that, while there was no evidence of a reduction in sensitivity to negative feedback after an MDmc lesion, there was a specific change in how positive feedback influenced future choices in the post-reversal period." However, the statistical test presented just prior to this statement showed that the interaction term including pre- and post-reversal was not significant.

The reviewer is quite right to point this out and we’re sorry that we were not clearer here. While it is correct that the interaction term with pre- and post-reversal was not significant, there was a strong trend to significance (*p* = 0.054). Given the effects on overall choice behaviour pre- and post-reversal, we therefore performed separate repeated measures ANOVAs on the pre-reversal and post-reversal data, which revealed a significant lesion group x surgery x previous outcome interaction in the post-reversal period. In the revised manuscript, we have outlined our analysis steps more clearly.

“In fact, as can be observed in Figure 4, the rate of switching post-reversal actually *increased* in the MDmc group after surgery.

[…]

Further post-hoc tests showed that this effect was mainly driven by a selective increase in the MDmc group in a tendency to switch to choosing a different stimulus just after having received a reward (increase in switching probability from pre- to post-MD surgery: 0.21 ± 0.07, *p* = 0.056), an effect less evident after no reward (0.06 ± 0.08, *p* = 0.47) or in the control animals after either outcome (reward: –0.04 ± 0.06, *p* = 0.56; no reward 0.02 ± 0.05 *p* = 0.74).”

We have also qualified the statements describing these effects. Therefore, rather than saying there was a “specific change in how positive feedback influenced future choices in the post-reversal period”, we now state:

“This change in switching behaviour highlights that, in addition to the absence of evidence for a reduction in sensitivity to negative feedback after an MDmc lesion, there was a change in how positive feedback influenced future choices, particularly in the post-reversal period. As can be seen in Figure 4, this meant that in the post-reversal phase, monkeys with MDmc damage became no more likely to stay with a current choice after reward delivery than reward omission.”

Finally, the regression described in Figure 6 should be described better. What exactly are the "averages" shown in the grids in panel a? across monkeys? Does each square correspond to a single coefficient (if so, how?) or several (this is my guess – one for each pairing of three choices and two outcomes for the given n, correct?)?

We are not quite sure that we fully understand all of the questions the reviewer has posed here, but will try to clarify our approach below in the hope that this will address all of his/her concerns.

The logistic regression examined the influence on the subsequent choice of all possible combinations of the past 6 choices and past 6 rewards in each monkey, both pre- and post-MD surgery (see Figure 6 left panel – the regressors associated with the combinations of the 6^th^ choice / reward were omitted from the figures and analyses as the purpose of these regressors was to pick up longer term choice/reward trends rather than to capture recent learning). If the animals were just using the correct contingent learning mechanism – associating each choice with its contingent outcome – all the weight of influence should lie on the diagonal, marked with red crosses. However, we had previously observed that even normal monkeys also display influences of combinations of (i) choices and non-contingent rewards received in the past (e.g., choice on trial n-2 x outcome of trial n-1: green area on the matrix) and (ii) past rewards and non-contingent choices made in subsequent trials (e.g., choice on trial n-1 x outcome of trial n-2: blue area on the matrix).

To calculate the regression weights, in each animal we performed 3 separate logistic regression analyses for each of the 3 potential stimuli (A, B, C). For each individual regression, the stimulus in question (e.g., ‘A’) would take the value of 1 whenever chosen and 0 whenever one of the other two stimuli (e.g., ‘B’ or ‘C’) was chosen. We then formed explanatory variables (EVs) based on all possible combinations of recent past choices and recent past rewards (trials n-1, n-2, …, n-6). Each EV took the value of 1 when, for the particular choice-outcome interaction, the monkey chosen A and was rewarded, –1 when the monkey chose B or C and was rewarded, and the 0 when there was no reward. We then fit a standard logistic regression with these 36 EVs (25 EVs of interest and 11 additional confound regressors describing combinations of choice / outcome n-6). This gave us estimates of β^Aand C^A. We then repeated this process for the other two stimuli to give us 3 sets of regression weights, β^A,β^B,β^C and three sets of covariances C^A,C^B,C^C. The regression weights into a single weight vector using a variance-weighted mean:β^=(C^A−1+C^B−1+C^C−1)−1(C^A−1β^A+C^B−1β^B+C^C−1β^C)